



**Tropical tropospheric ozone distribution and trends from in situ and satellite**
**data**
**Audrey Gaudel[1,2*], Ilann Bourgeois[1,2,#], Meng Li[1,2], Kai-Lan Chang[1,2], Jerald Ziemke[3,4],**
**Bastien Sauvage[5], Ryan M. Stauffer[3], Anne M. Thompson[3,6], Debra E. Kollonige[3,7], Nadia**
**Smith[8], Daan Hubert[9], Arno Keppens[9], Juan Cuesta[10], Klaus-Peter Heue[11], Pepijn**
**Veefkind[12,13], Kenneth Aikin[1,2], Jeff Peischl[1,2], Chelsea R. Thompson[2], Thomas B. Ryerson[2],**
**Gregory J. Frost[2], Brian C. McDonald[2], Owen R. Cooper[1,2]**
**[1]CIRES, University of Colorado, Boulder, USA**
**[2]NOAA Chemical Sciences Laboratory, Boulder, USA**
**[3]NASA Goddard Space Flight Center, Greenbelt, Maryland, USA**
**[4]Morgan State University, Baltimore, Maryland, USA**
**[5]Laboratoire d'Aérologie, Université de Toulouse, CNRS, Université Toulouse III Paul**
**Sabatier, France**
**[6]University of Maryland Baltimore County, Baltimore, MD, USA**
**[7]Science Systems and Applications, Inc., Lanham, MD, USA**
**[8]Science and Technology Corporation, Madison, Wisconsin, 53703, USA**
**[9]Royal Belgian Institute for Space Aeronomy (BIRA-IASB), 1180 Brussels, Belgium**
**[10]Laboratoire Inter-universitaire des Systèmes Atmosphériques (LISA), UMR7583,**
**Universités Paris-Est Créteil et Paris, Diderot, CNRS, Créteil, FR**
**[11]Technische Universität München (TUM), School of Engineering and design, and**
**Deutsches Zentrum für Luft- und Raumfahrt (DLR), Institut für Methodik der**
**Fernerkundung (IMF), Oberpfaffenhofen, Germany**
**[12]Royal Netherlands Meteorological Institute, De Bilt, 3731 GA, the Netherlands**
**[13]Faculty of Civil Engineering and Geosciences, University of Technology Delft, Delft, 2628**
**CN, the Netherlands**
**[#]Now at University Savoie Mont Blanc, INRAE, CARRTEL, F-74200 Thonon-les-Bains,**
**France**
**\*Correspondence: audrey.gaudel@noaa.gov**





**Abstract**

Tropical tropospheric ozone (TTO) is important for the global radiation budget because the longwave radiative effect of tropospheric ozone is higher in the tropics than mid-latitudes. In recent decades the TTO burden has increased, partly due to the ongoing shift of ozone precursor emissions from mid-latitude regions toward the equator. In this study, we assess the distribution and trends of TTO using ozone profiles measured by high quality in situ instruments from the IAGOS (In-Service Aircraft for a Global Observing System) commercial aircraft, the SHADOZ (Southern Hemisphere ADditional OZonesondes) network, and the ATom (Atmospheric Tomographic Mission) aircraft campaign, as well as six satellite records reporting tropical tropospheric column ozone (TTCO): TROPOMI, OMI, OMI/MLS, OMPS/MERRA2, CrIS, and IASI/GOME2. With greater availability of ozone profiles across the tropics we can now demonstrate that tropical India is among the most polluted regions (e.g., Western Africa, tropical South Atlantic, Southeast Asia, Malaysia/Indonesia) with present-day $95^{th}$ percentile ozone values reaching 80 nmol mol$^{-1}$ in the lower free troposphere, comparable to mid-latitude regions such as Northeast China/Korea. In situ observations show that TTO increased between 1994 and 2019, with the largest mid- and upper tropospheric increases above India, Southeast Asia and Malaysia/Indonesia (from $3.4 \pm 0.8$ to $6.8 \pm 1.8$ nmol mol$^{-1}$ decade$^{-1}$), reaching $11 \pm 2.4$ and $8 \pm 0.8$ nmol mol$^{-1}$ decade$^{-1}$ close to the surface (India and Malaysia/Indonesia, respectively). The longest continuous satellite records only span 2004-2019, but also show increasing ozone across the tropics when their full sampling is considered, with maximum trends over Southeast Asia of $2.31 \pm 1.34$ nmol mol$^{-1}$ decade$^{-1}$ (OMI) and $1.69 \pm 0.89$ nmol mol$^{-1}$ decade$^{-1}$ (OMI/MLS). In general, the sparsely sampled aircraft and ozonesonde records do not detect the 2004-2019 ozone increase, which could be due to the genuine trends on this timescale being masked by the additional uncertainty resulting from sparse sampling. The fact that the sign of the trends detected with satellite records changes above three IAGOS regions, when their sampling frequency is limited to that of the in situ observations, demonstrates the limitations of sparse in situ sampling strategies. This study exposes the need to maintain and develop high frequency continuous observations (in situ and remote sensing) above the tropical Pacific Ocean, the Indian Ocean, Western Africa and South Asia in order to estimate accurate and precise ozone trends for these regions. In contrast, Southeast Asia and Malaysia/Indonesia are regions with such strong increases of ozone that the current in situ sampling frequency is adequate to detect the trends on a relatively short 15-year time scale.

**Plain Language Summary**

Tropospheric ozone is an air pollutant and a climate forcer, and plays an important role in the global Earth's radiation budget, especially in the tropics. In recent decades, the tropical tropospheric ozone burden has increased, partly due to the ongoing shift of ozone precursor emissions from mid-latitudes toward the equator. In this study, we assess the changes in time of tropical tropospheric ozone using in situ ozone profiles measured by high quality instruments from commercial aircraft, ozonesondes and satellites. In situ observations show that tropical tropospheric ozone increased between 1994 and 2019, with the largest increases above India, Southeast Asia and Malaysia/Indonesia. The longest continuous satellite records of ozone only span 2004-2019, but show increasing ozone across the tropics, with maximum trends over





Southeast Asia. In general, the sparsely sampled aircraft and ozonesonde records do not detect the 2004-2019 ozone increase, which could be due to sample sizes that are too small for accurate trend detection on this relatively short 15-year time period. The fact that the satellite records also fail to consistently detect positive trends when their sampling frequency is limited to that of the in situ observations demonstrates the limitations of sparse in situ sampling in the tropics. This study demonstrates the need to maintain and develop continuous observations (in situ and remote sensing) above the tropical Pacific Ocean, the Indian Ocean, Western Africa and South Asia in order to estimate accurate and precise ozone trends for these regions.

**Short Summary (500 characters)**
The study examines tropical tropospheric ozone changes. In situ data from 1994-2019 display increased ozone, notably over India, Southeast Asia, and Malaysia/Indonesia. Sparse in situ data limit trend detection for the 15-year period. In situ and satellite data, with limited sampling, struggle to consistently detect trends. Continuous observations are vital over the tropical Pacific Ocean, Indian Ocean, Western Africa, and South Asia for accurate ozone trend estimation in these regions.

**1. Introduction**

Tropospheric ozone negatively affects human health and vegetation, and it is a short-lived climate forcer (Fleming et al., 2018; Mills et al., 2018; Gulev et al., 2022; Szopa et al., 2022). The longwave radiative effect of tropospheric ozone is higher in the tropics and subtropics (between 30˚S and 30˚N) compared to mid-latitudes (Doniki et al., 2015; Gaudel et al., 2018). The most recent IPCC assessment concluded with a high level of confidence that tropical ozone increased by 2-17% per decade in the lower troposphere, and by 2-12% per decade in the free troposphere from the mid-1990s to the period 2015-2018 (Gulev et al., 2021). These increases are especially strong across southern Asia (Gaudel et al., 2020), and according to the longest available satellite record, ozone increases in this region have been occurring since at least 1979 (Ziemke et al., 2019). A comprehensive NASA analysis used the OMI/MLS satellite record to show a clear increase of tropospheric column ozone (1-2.5 DU decade$^{-1}$) between 2005 and 2016 throughout the tropics, with larger trends over the Arabian Peninsula, India and Southeast Asia, generally consistent with a simulation by NASA's MERRA-2 GMI global atmospheric chemistry model (Ziemke et al., 2019). Similar results were found in a recent study using the NASA Goddard Earth Observing System Chemistry Climate Model (Liu et al., 2022). Weak to moderate positive trends of 0.6 and 1.5 nmol mol$^{-1}$ decade$^{-1}$ between 1995 and 2015-2018 were also reported at two remote tropical surface sites (Mauna Loa, Hawaii, and American Samoa, South Pacific; Cooper et al., 2020). A recent analysis of 1998-2019 tropical ozone trends using the Southern Hemisphere ADditional OZonesondes (SHADOZ) network reported highly



seasonal but overall weak positive trends (1-2% decade$^{-1}$) in the mid-troposphere (5-10 km)
(Thompson et al., 2021).
Simulations by a wide range of global atmospheric chemistry models show that global-
scale increases of tropospheric ozone since pre-industrial times are driven by anthropogenic
emissions of ozone precursor gases (Archibald et al., 2020; Skeie et al., 2020; Griffiths et al.,
2021; Szopa et al., 2021; Wang et al., 2022; Fiore et al. 2022), with approximately 54% of the
1850-2000 global tropospheric ozone increase occurring in the tropics (30° S – 30° N) (Young et
al., 2013).  A key ozone precursor that drives the background increase of tropospheric ozone,
especially in the free troposphere is methane (Thompson and Cicerone, 1986a,b; Hogan et al.,
1991; Fiore et al., 2002).  From 1980 to 2010 the estimated increase of the global tropospheric
ozone burden due to the increase of anthropogenic emissions and the partial shift of the
emissions from mid-latitudes towards the equator was 28.12 Tg (8.9%), with the increase of
methane (15%) accounting for one quarter of the ozone burden increase (as simulated by the
CAM-chem model; Zhang et al., 2016). Most of the ozone burden increase (64%) occurred in the
tropics (30° S – 30° N), driven by emissions from South Asia, Southeast Asia and by increasing
background methane levels (Zhang et al., 2021).  Similar rates of ozone burden increases,
peaking in the tropics, are simulated by a range of CMIP6 models (1995-2014) (Skeie et al.,
2020,) the GEOS-Chem model (1995-2017) (Wang et al., 2022), the JPL TCR-2 chemical
reanalysis (1995-2018) (Miyazaki et al., 2020), and a 15-member initial-condition ensemble
generated from the CESM2-WACCM6 chemistry-climate model (1950-2014) (Fiore et al.,
2022). The increase of methane has continued to the present and the observed global mean
methane increase from 1983 to 2023 is 18% (the increase is 8% since 2004 when the OMI
satellite instrument began operations) (www.gml.noaa.gov).  Under a future scenario of high
anthropogenic emissions and continuously increasing methane concentrations (Griffiths et al.,
2021), the global ozone burden is expected to increase for the remainder of the 21$^{st}$ century (see
the ssp370 scenario in Figure 6.4 of Szopa et al., 2021), with increases of approximately 10%
from 2014 to 2050. In the tropics the strongest increases (though 2050) are expected across
South Asia (10-20%), with little or no increase across the remote regions of the equatorial Pacific
and equatorial Atlantic.
The tropics are characterized by high ozone values over the southern tropical Atlantic and
Southeast Asia (Fishman et al., 1990; Fishman et al., 1996; Thompson et al., 1996; Logan et al.,
1999; Ziemke et al., 2019) and low ozone values (< 10 nmol mol$^{-1}$) in the free troposphere over
the Pacific warm pool (Kley et al., 1996), although these low values have become less frequent
over the last two decades (Gaudel et al., 2020). The spatial distribution of tropical tropospheric
ozone (TTO) can vary on a range of timescales. On multi-year timescales TTO experiences a
dipole oscillation across the tropical Pacific Ocean due to El Niño-Southern Oscillation (ENSO)
(Chandra et al., 1998; Doherty et al., 2006; Oman et al., 2013; Xue et al., 2020). On seasonal
time scales ozone can vary with the Madden-Julian Oscillation (MJO) (Ziemke et al., 2015), and
also with dry and wet conditions (a.k.a. biomass burning and monsoon seasons) related to the
seasonal shifts of the Intertropical Convergence Zone (ITCZ) (Fishman et al., 1992; Oltmans et
al., 2001; Sauvage et al., 2007; Thompson et al., 2012). In a given season, TTO can be further
influenced by biomass burning, lightning, inter-hemispheric transport and stratospheric
intrusions/large-scale subsidence (Sauvage et al., 2007; Jenkins et al., 2014; Yamasoe et al.,



2015; Hubert et al., 2021). For instance, high ozone concentrations were recently measured above the tropical Atlantic (Bourgeois et al., 2020), and were attributed to biomass burning emissions, whose effects on tropospheric ozone enhancements are underestimated by global chemistry-transport models, especially in the tropics and the southern hemisphere (Bourgeois et al., 2021).

While decades of research on the distribution of TTO using satellite instruments (Fishman et al., 1986, 1987, 1990; Ziemke et al., 1998, 2005, 2009, 2011, 2019) and in situ observations (Logan et al., 1999; Thompson et al., 2000, 2003, 2012, 2021; Oltmans et al., 2001; Sauvage et al., 2005; Sauvage et al., 2007; Yamasoe et al., 2015; Tarasick et al., 2019; Cooper et al., 2020, Lannuque et al., 2021) have characterized the spatial and temporal variability of TTO concentrations, reconciling differences between satellite and in situ observations has been a challenge (Gaudel et al., 2018).

To update our understanding of tropospheric ozone's distribution and trends across the tropics, this study presents a quantitative analysis of four complementary data sets in time and space across the 20˚S-20˚N latitude band: (1) Thousands of vertical ozone profiles from the In-Service Aircraft for a Global Observing System (IAGOS) (Nédélec et al., 2015; Blot et al., 2021) above five continental regions; (2) Regular vertical profiles from the SHADOZ ozonesonde network (Thompson et al., 2017; Stauffer et al., 2022) above 14 continental and oceanic sites; (3) Vertical profiles from the Atmospheric Tomographic Mission (ATom) aircraft campaign above five oceanic regions; (4) Tropospheric column ozone retrievals from four well-known and two new satellite records.

The paper is organized as follows. Section 2 describes the data sets and the methodology for quantifying the distribution and trends of ozone. Section 3 presents the results that include the distribution of ozone from the in situ data, an evaluation of the satellite records and the trend estimates from IAGOS, SHADOZ and satellite records. Section 4 presents the main conclusions.

**2. Methods**

We define the tropics as the latitude band between 20˚S and 20˚N, within the bounds of the Tropic of Cancer and the Tropic of Capricorn. This latitude band covers most of the Southern Hemisphere ADditional OZonesondes (SHADOZ) network designed to measure ozone in the subtropics/tropics. The goal of the study is to characterize the 20˚S-20˚N latitude band that can be impacted by subtropical air masses in some regions, especially at the edge of the domain. The satellite data are shown for the same domain but we also include one satellite record, for which the tropical tropospheric column ozone (TTCO) retrieval is based on the cloud slicing technique, that is limited to 15˚N -15˚S.

We focus on three time periods: 2014-2019, also called "present-day" to assess the distribution of TTO (5th, 50th, and 95th percentiles) with in situ data above the sampled regions and sites described in Figure 1; 1994-2019 to assess ozone trends using in situ data records for more than two decades; 2004-2019 to assess ozone trends over the time period of the Ozone Monitoring Instrument (OMI) data set, which is the longest time series of ozone measured from space from a satellite.

We also use new datasets to assess the distribution of TTO, such as the ATom aircraft campaign, and the CrIS and IASI/GOME2 satellite records.



**2.1 In situ measurements**

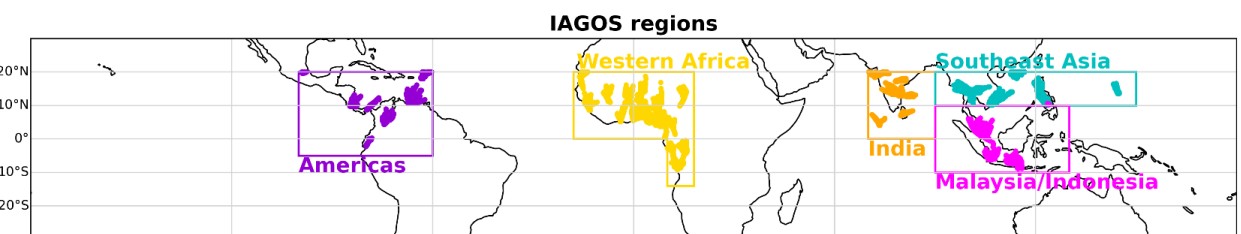

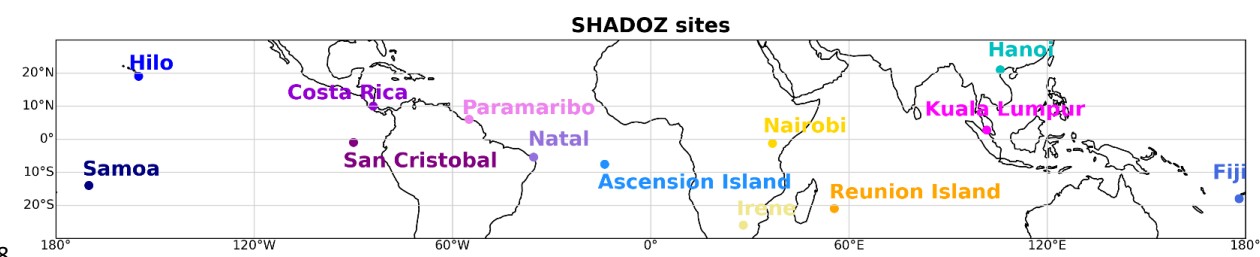

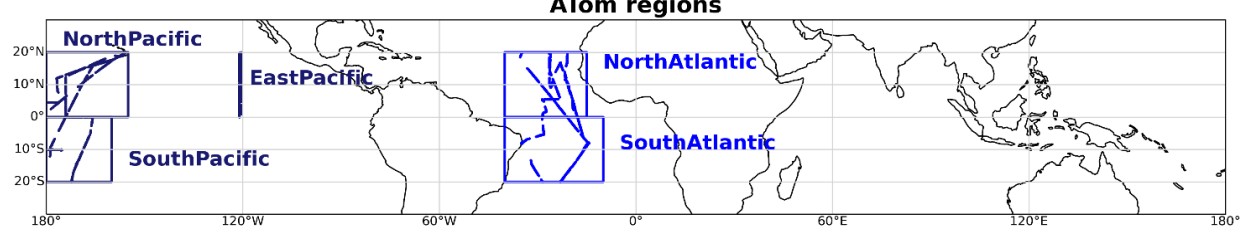

**Figure 1**. Regions and sites of IAGOS, SHADOZ and ATom measurements used in this study to
assess the 5th, 50th and 95th percentiles of ozone in the tropical troposphere over 2014-2019. Data
from IAGOS and ATom flights are clustered into specific regions such as Americas, Africa,
South Asia, Southeast Asia, Malaysia/Indonesia, North Pacific, South Pacific, East Pacific,
North Atlantic and South Atlantic. IAGOS and ATom flight tracks are plotted on the map to
show the specific sampling locations for 2014-2019. IAGOS and SHADOZ data are statistically
fused above the Americas, Southeast Asia and Malaysia/Indonesia and used to estimate ozone
trends between 1994 and 2019. For India, only IAGOS data are available for the ozone trend
estimate between 1994 and 2019.

**2.1.1 IAGOS**
Description: The European research infrastructure In-service Aircraft for a Global
Observing System (IAGOS), formerly known as the Measurement of Ozone and Water Vapor by
Airbus In Service Aircraft (MOZAIC), has collected continuous high quality ozone profiles up to



12 km (~ 200 hPa) on-board commercial aircraft since 1994 (Blot et al., 2020). Ozone is
measured using a UV analyzer (Thermo Scientific, model 49) and the total uncertainty is ±2
nmol mol$^{-1}$ ±2% (Nédélec et al., 2015).

Data treatment: For this study, we consider five tropical regions: Americas, Africa, India,

Southeast Asia and Malaysia/Indonesia. We use IAGOS data to assess the average ozone
distribution between 2014 and 2019, referred to as "present-day ozone", as well as to assess
ozone trends between 1994 and 2019. Over the time period 1994-2019, the most frequented
airports were Caracas (1214 profiles) and Bogota (560 profiles) for the Americas; Lagos (761
profiles) and other airports in the Gulf of Guinea for Western Africa; Chennai (680 profiles) and
Hyderabad (552 profiles) for India; Bangkok (1535 profiles) and Ho Chi Minh City (367
profiles) for Southeast Asia; Singapore (265 profiles), Kuala Lumpur (208 profiles) and Jakarta
(113 profiles) for Malaysia/Indonesia (Table S1). All available ozone profiles from these airports
are used in this study. The individual ozone profiles are averaged to a common vertical
resolution of 10 hPa prior to any further analysis. To assess the annual ozone distribution the
profiles are averaged annually. To assess ozone trends, the quantile regression method is applied
to individual profiles (section 2.5). To compare with the satellite data, the profiles were averaged
monthly before being converted to a tropospheric column value ranging from the surface up to
270 hPa or up to the maximum altitude (~ 200 hPa). We chose 270 hPa to be consistent with the
TROPOMI tropical tropospheric column ozone. While some of the satellite records used in this
study have an upper limit at 150 hPa (thermal tropopause), IAGOS commercial aircraft do not
reach these altitudes.

**2.1.2 SHADOZ**

Description: The Southern Hemisphere ADditional OZonesondes (SHADOZ) network

has provided ozone profiles at multiple sites between 25ºS and 21ºN since 1998, and presently
operates 14 sites. SHADOZ is a NASA-sponsored project operated by NOAA and 15 institutions
around the world (Thompson et al., 2003a, 2003b, 2012, 2021). The SHADOZ archive of ozone
profiles, measured by electrochemical concentration cell (ECC) ozonesondes, were reprocessed
in 2016-2018 (Witte et al., 2017; 2018). In comparisons of the reprocessed data with collocated
total ozone spectrometers and satellite overpasses, the reprocessed SHADOZ total ozone column
(TOC) agreed with the independent data to 2% (Thompson et al., 2017). SHADOZ data since
2018 have been collected and processed according to the same protocols as the reprocessed
profiles (Stauffer et al., 2018, 2020; 2022; WMO/GAW 268, 2021). A recent study of TOC
stability over 60 global stations revealed an artifact of declining tropospheric ozone at the
SHADOZ Hilo and Costa Rican stations (Stauffer et al., 2020; 2022). Those data were not used
in the recent Thompson et al. (2021) study that showed distinctive seasonal and regional
variations in ozone trends collected at eight SHADOZ stations within ±15º latitude of the
equator.

Data treatment: As with the IAGOS data, the SHADOZ ozone profiles were averaged to

a common vertical resolution of 10 hPa before any further analysis. The 10 hPa-resolution
vertical profiles are fused with the IAGOS 10 hPa-resolution vertical profiles to assess trends
between the surface and 200 hPa (section 2.6). To compare with the satellite data, the profiles



were averaged monthly before being converted to tropospheric columns up to 270 hPa, 150 hPa
and 100 hPa.

**2.1.3 ATom**
Description: The Atmospheric Tomography (ATom) project was a global scale NASA
aircraft mission which collected profiles of ozone and hundreds of other atmospheric constituents
in remote regions above the Atlantic and Pacific basins on board the NASA DC-8 aircraft. The
project consisted of four seasonal circumnavigations of the globe, one in each season,
continually profiling the troposphere between 180 m and 14 km a.s.l. with a temporal resolution
of 10 Hz, averaged to 1 Hz (data available at https://espo.nasa.gov/atom, last access March 7,
2022). The ATom mission occurred in July–August 2016 (ATom-1), January–February 2017
(ATom-2), September– October 2017 (ATom-3), and April–May 2018 (ATom-4). Ozone was
measured using the National Oceanic and Atmospheric Administration (NOAA) nitrogen oxides
and ozone (NOyO3) instrument (Bourgeois et al. 2020). The total estimated uncertainty at sea
level is ± (0.015 nmol mol$^{-1}$ ± 2 %).
Data treatment: We used the ATom ozone profiles available above five regions in the
tropics:  North Pacific, South Pacific, East Pacific, North Atlantic and South Atlantic. Most of
the regions were sampled over one day in August 2016, February and October 2017, and May
2018, except the East Pacific which was sampled in July 2016, January and September 2017, and
April 2018. Each flight produced 6-14 profiles in each region. Therefore, the ATom dataset is
used to assess the ozone distribution over the 2016-2018 time-period and for the annual
comparison with the satellite products. As for IAGOS and SHADOZ, we averaged the profiles to
a common vertical resolution of 10 hPa within the five ATom regions. To compare with satellite
data, the profiles were converted to tropospheric column ozone from the near-surface
measurements up to 270 hPa and averaged for the entire ATom period above each of the five
regions.

**2.2 Tropical Tropospheric Column Ozone (TTCO) estimation from IAGOS, SHADOZ and**
**ATom**
In this study and as mentioned in Section 2.1, the ozone profiles from in situ observations
have been converted to columns to evaluate the satellite products. The current TOAR-II
Harmonization and Evaluation of Ground-based Instrument for Free Tropospheric Ozone
Measurements (HEGIFTOM) focus working group (https://hegiftom.meteo.be/) recommended
150 hPa as the top limit of the TTCO in the 15˚S-15˚N tropical band and 200 hPa in the 15˚S-
30˚S/15˚N-30˚N bands. As we focus our study on the 20˚S-20˚N latitude band, we decided to use
the 150 hPa top limit. Some variations on the TTCO definition occur in this study and are
detailed below, but are not corrected for.
IAGOS aircraft cannot reach 150 hPa as they have a maximum cruise altitude around 200
hPa. Therefore, only SHADOZ ozonesondes, which reach the mid- or upper stratosphere, were
used to calculate TTCO from the surface to 150 hPa. However, we additionally calculated TTCO
up to 270 hPa with IAGOS and ATom to compare with TROPOspheric Monitoring Instrument
(TROPOMI) and Infrared Atmospheric Sounding Interferometer (IASI) / Global Ozone
Monitoring Experiment 2 (GOME2) satellite data.





### 2.3 Satellite data

In this study we mainly focus on satellite data based on ultraviolet absorption (UV) retrievals, supplemented with two ozone records derived from infrared (IR) measurements as described below. Two key parameters differ between the satellite datasets: (i) the top limit used to define the tropospheric column ozone, and (ii) the horizontal coverage. Figure S1 shows the time series of the pressure level characterizing the top limit. Depending on the datasets, the top limit is constant or varies with time. The tropical coverage is 20˚S-20˚N for all satellite records except the Ozone Monitoring Instrument (OMI) data, which is constrained to 15˚S-15˚N. All satellite records were averaged to a common 5˚x5˚ monthly grid.

### 2.3.1 TROPOMI CCD

The TROPOspheric Monitoring Instrument (TROPOMI, Veefkind et al., 2012) was launched onboard the Sentinel-5 Precursor (S5P) satellite in October 2017. The tropospheric column ozone data from TROPOMI, inferred using the convective cloud differential technique (CCD, Ziemke et al., 1998; Heue et al., 2016; Hubert et al., 2021), covers the 20˚S-20˚N latitude band, between the surface and 270 hPa. For this study, we compute monthly data from daily measurements on a 5˚ x 5˚ grid to be consistent with the other satellite data records. For the 5˚ x 5˚ gridded data we estimate the uncertainty of the TROPOMI CCD tropospheric ozone column to be about 2 DU. We only use data from 2019, which is the last year of our present-day time period 2014-2019.

### 2.3.2 OMI CCD

The Ozone Monitoring Instrument (OMI) was launched onboard the Aura satellite in July 2004. For this study we used tropical tropospheric column ozone retrieved using the CCD technique (Ziemke et al., 1998; Ziemke and Chandra, 2012), which is consistent with TROPOMI-derived TTCO. The tropospheric column is defined between the surface and 100 hPa, and it is constrained in the 15˚S-15˚N latitude band inherent to the CCD technique. OMI records are available since 2004 and for this study we use monthly means to assess ozone distribution during the present-day time period of 2014-2019 as well as the trends of ozone over 2004-2019. The monthly accuracy and precision (1σ) are 3 and 3.5 DU, respectively.

### 2.3.3 OMI/MLS

The OMI and the Microwave Limb Sounder (MLS) sensors are both onboard the Aura satellite and the tropospheric column ozone is retrieved by subtracting the stratospheric column ozone measured by MLS from the total column ozone measured by OMI (Ziemke et al., 2006). The top limit of the OMI/MLS tropospheric column ozone is the thermal tropopause calculated from NCEP reanalysis data using the World Meteorological Organization (WMO) 2 K km$^{-1}$ lapse-rate definition. The tropopause varies seasonally between 95 and 115 hPa (Figure S1). OMI/MLS data cover the 60˚S-60˚N latitude band and for this study we focus on the 20˚S-20˚N latitude band. The monthly accuracy and precision (1σ) are 2 and 1.5 DU, respectively. Further details of the OMI/MLS product and a description of an updated drift correction can be found in Section S.2 of the supplementary material.




### 2.3.4 OMPS/MERRA2

The Ozone Mapping Profiler Suite (OMPS) was launched in January 2012 onboard the
Suomi National Polar-orbiting Partnership (Suomi NPP) spacecraft. The tropospheric column
ozone is retrieved by subtracting the stratospheric column of MERRA2 (Modern-Era
Retrospective analysis for Research and Applications, version 2) ozone reanalysis data from the
total column ozone of the OMPS nadir mapper (Ziemke et al., 2019). The derived daily
tropospheric column ozone uses the MERRA2 tropopause with assimilated MLS ozone. The
MERRA2 tropopause was determined using a potential vorticity (PV) – potential temperature (θ)
definition (2.5 PV units, 380 K; Wargan et al., 2020). The tropopause at a given grid point was
taken as the larger of these two PV and θ surfaces. However, in this study, the tropopause is
exclusively defined by θ surfaces as we focus on the 20˚S-20˚N latitude band. For the MERRA2
assimilation, in 2015 MLS changed from version 2.2 to version 4.2 (Wargan et al., 2017; Davis
et al., 2017). This produced a 1-1.5 DU difference between the earlier and latter record for
stratospheric column ozone, which prevents accurate trend detection from either MERRA2
stratospheric column ozone or the derived tropospheric column ozone from OMPS/MERRA2.
The OMPS/MERRA 2 tropopause pressure varies seasonally between 95 hPa and 108 hPa
(Figure S1). The monthly accuracy and precision (1σ) are 3 and 2 DU, respectively.

### 2.3.5 CrIS

The Cross track Infrared Sounder (CrIS) is onboard the Suomi NPP (2011–2021) and
JPSS-1 (NOAA-20 in operations; 2017–present) and builds upon the hyperspectral IR record
first started by the Atmospheric Infrared Sounder (AIRS) on Aqua (2002–2022). For this study
we are focusing on the ozone profiles retrieved by the Community Long-term Infrared
Microwave Combined Atmospheric Product System (CLIMCAPS, Smith and Barnet, 2019;
2020). CLIMCAPS retrieves atmospheric state parameters, including ozone profiles (from the
surface to the top of the atmosphere), from AIRS and CrIS to form a long-term record that spans
instrument and platform differences. CLIMCAPS uses MERRA2 as the a-priori for ozone. Here
we focus on CLIMCAPS from CrIS onboard Suomi NPP (National Polar-orbiting Partnership,
2016-01-01 to 2018-03-31) and NOAA-20 (previously known as JPSS-1, 2018-04-01 to 2022-
08-31) for the time period 2016-2019 because this gives us the baseline IR sounding capability
for the next two decades (CrIS is scheduled for launch on three additional JPSS platforms). CrIS
data covers the 90˚S-90˚N latitude band and for this study we focus on the 20˚S-20˚N latitude
band. The accuracy that CrIS vary between -9.4% globally and -20% in the tropics compared
with ozonesondes. The precision that CrIS globally is 21.2% (Nalli et al., 2017).
For CrIS, we accessed CLIMCAPS Level 2 retrievals via NASA GES DISC (NASA
Goddard Earth Sciences Data and Information Services Center; Sounder SIPS, & Barnet, Chris.,
2020a and 2020b; https://disc.gsfc.nasa.gov/). We aggregated them onto 1˚ equal angle global
grids. Specifically, we accessed the ozone retrieved fields (o3_mol_lay) defined as 100 layer
column density profiles [molec m$^{-2}$] and subset them into tropospheric profiles. We defined the
troposphere as all values between Earth surface (prior_surf_pres) and tropopause (tpause_pres).
A total column value is simply the sum of all column density values, converted to DU. We used
the quality flag (ispare_2=0) to define all successful retrievals, which we simply averaged per



grid box. No other filtering was done. CLIMCAPS retrievals are done from cloud cleared
radiances so we do not have to make specific accommodation for clouds.
**2.3.6 IASI / GOME2**
IASI/GOME2 is a multispectral approach used to retrieve ozone for several partial
columns. It is based on the synergism of IASI and GOME-2 measurements respectively in the
thermal infrared and the ultraviolet spectral domain, jointly used in terms of radiance spectra for
enhancing the sensitivity of the retrieval for lowermost tropospheric ozone (below 3 km above
sea level, see Cuesta et al., 2013). Studies over Europe and East Asia have shown good skill for
capturing near surface ozone variability compared to surface in situ measurements of ozone
(Cuesta et al. 2018; 2022). This ozone product offers global coverage for low cloud fraction
conditions (below 30%) for 12-km diameter pixels spaced by 25 km (at nadir pointing). The
IASI/GOME2 global dataset is publicly available through the AERIS French data center, with
data from 2017 to the present (available at https://iasi.aeris-data.fr/o3_iago2/, last accessed
08/02/2023) and covers the 90˚S-90˚N latitude band. For this study, we are using the 2017-2021
monthly tropospheric column ozone between the surface and 12 km, focusing on the 20˚S-20˚N
latitude band.
**2.4 Comparison between satellite and in situ data**
To assess the performance of the six satellite records, we calculated the mean biases
between satellite-detected monthly TTCO and IAGOS and SHADOZ integrated profiles over the
2014-2019 time period. The biases are calculated as follows:
$$\text{Mean Bias (MB in DU)} = \frac{\sum_{i=1}^{N} y_{i(sat)} - y_{i(ref)}}{N}$$

$$\text{Normalized Mean Bias (NMB in \%)} = \frac{1}{N}\sum_{i=1}^{N} \frac{y_{i(sat)} - y_{i(ref)}}{y_{i(ref)}} \times 100$$

N is the number of monthly TTCO observations over a given region/site and $y_i$ is the
monthly mean TTCO based on in situ data (ref) or satellite data (sat).
In order to represent the relationship between the satellite data and the in situ data, we
used a least-square linear regression as well as the orthogonal distance regression (ODR). In this
exercise, we are not using strict sampling criteria in time and space (except for the satellite and
in-situ observations being in the same month, year and grid cell), nor smoothing in situ ozone
profiles to the vertical resolution of the satellite data before integration. To extract satellite data
over IAGOS and ATom regions, we used a 5˚x5˚ gridded mask reflecting monthly grid cells with
available IAGOS and ATom data, and only these grids are used to compute regional mean
satellite values. For comparison to SHADOZ data, satellite data were extracted at the latitude and
longitude of the SHADOZ sites (sonde launch site within satellite pixel). We include all satellite
records with a minimum of one year of data within 2014-2019.
**2.5 Fused product and trend estimation**



The tropical region has sparse in situ sampling in both time and space, which makes accurate quantification of trends challenging. Based on a sampling sensitivity test (section S1, Figures S2 and S3), we conclude that one profile per week is only sufficient for detection of trends with a very strong magnitude (i.e., $> |3|$ nmol mol$^{-1}$ decade$^{-1}$), which is not common in the free troposphere. We show that a sampling frequency of 7 profiles per month is sufficient for basic trend detection (i.e., to reliably determine if there is a trend) of TTO using the datasets presently available (if the magnitude of a trend is greater than $|1|$ nmol mol$^{-1}$ decade$^{-1}$), but additional data are required for accurate quantification or detection of a weaker trend.

Because the sparse sampling makes trend detection difficult, we have chosen to statistically fuse the in situ measurements from the IAGOS and SHADOZ programs over large regions, which includes air masses from different origins and influences (Figures 1 and S7 to S11). The method is based on a data fusion technique described by Chang et al. (2022), which considers ozone correlation structure, sampling frequency and inherent data uncertainty. By investigating systematic ozone variability, the resulting fused product allows us to reconcile the differences between heterogeneous datasets and enhance the detectability of trends. For the Americas, we fused SHADOZ data over San Cristobal and Paramaribo with the IAGOS data (Figure S7); for Southeast Asia, we fused SHADOZ over Hanoi with the IAGOS data (Figure S8); for Malaysia/Indonesia, we fused SHADOZ data over Kuala Lumpur and Watukosek (Java) with the IAGOS data (Figure S9). For Western Africa and India, SHADOZ data are not available and we show the timeseries of just the IAGOS data in Figure S10 and S11, respectively.

For IAGOS data and the fused product, the trend estimate and its associated uncertainty are based on quantile regression (Koenker & Hallock, 2001), which is an appropriate choice for ozone profile time series, because of the irregular sampling schemes and the need to evaluate ozone changes associated with a range of percentiles (Chang et al., 2021). Data gaps are not interpolated as interpolation creates fictitious sample sizes for trend detection, while treating the missing data as not substantially deviant from the available data variability. Due to limited available sample sizes, only median trends (i.e., an estimate of the trend based on median data values) are reported in this study. To account for potential correlation between ozone and climate variability, such as ENSO (El Niño-Southern Oscillation) and QBO (quasi-biennial oscillation), the trend model is specified through:

$$\text{anomaly} = b0 + b1\ \text{Trend} + b2\ \text{ENSO} + b3\ \text{QBO(30mb)} + b4\ \text{QBO(50mb)} + \text{Noise} \quad [1]$$

where b0 is the intercept, b1 is the linear trend, b2 is the regression coefficient for ENSO, b3 and b4 are coefficients for QBO at 30 and 50 mb, respectively. The trend uncertainty is derived by a bootstrapping method (Feng et al., 2011). The ENSO and QBO indexes can be found in the data availability section. Figure S12 shows that if ENSO and QBO are not considered, the trends can be offset by about 1-2 nmol mol$^{-1}$ decade$^{-1}$ at individual pressure layers over the five IAGOS regions, except Africa where the trend differences are negligible.

In addition, we conducted trend analysis of the monthly TTCO from SHADOZ, IAGOS, OMI and OMI/MLS as well as the tropical ozone burden (TOB, Tg decade$^{-1}$) over zonal monthly means using OMI and OMI/MLS. The OMI/MLS TTCO has shown a drift over time that we corrected for this study (see section S2).





**3. Results**
**3.1 Ozone Profiles**

**Figure 2.** Distribution of TTO showing annual 50th, 5th and 95th percentiles (left, center, and right
columns, respectively) of ozone profiles (nmol mol⁻¹) measured by IAGOS (top), SHADOZ
(middle) both between 2014 and 2019, and ATom (bottom) between 2016 and 2018. The colors
correspond to the IAGOS, ATom regions and SHADOZ sites (see Figure 1). The North China
and Korea (NE_China_Korea) region from IAGOS data is plotted in grey on all panels as a
reference for mid-latitude polluted regions.





For the period 2014-2019 (IAGOS, SHADOZ) and 2016-2018 (ATom), the three in situ data sets
show a range of ozone values from the surface to 200 hPa, indicative of the different
photochemical and transport regimes across the tropics (Figure 2). Here we highlight several
notable features. The 50[th] and 95[th] percentiles of SHADOZ data over Hanoi (up to 100 nmol mol[-]
[1]) are much higher than at the other sites/regions, especially below 750 hPa. Hanoi experiences
strong regional ozone production with a significant contribution from biomass burning in the
Indochina peninsula, especially in spring (Ogino et al., 2022). Ozone is lowest above the tropical
South Pacific (dark blue lines on the SHADOZ and ATom panels of Figure 2) and the Americas
(purple lines on the IAGOS, SHADOZ panels of Figure 2) with the 5[th] percentile less than 10
nmol mol[-1], especially in the lower troposphere. The 95[th] percentile ozone is highest above
Africa, India and Southeast Asia in the mid- and upper troposphere, and above Southeast Asia
and Malaysia/Indonesia in the boundary layer.  The tropical South Atlantic (ATom and
Ascension Island) is also notable due to broad enhancements from the lower free troposphere to
the upper troposphere, with values of 60-80 nmol mol[-1]. Similar patterns are seen in the median
(50[th] percentile) ozone profiles, albeit with lower mixing ratios.
As a frame of reference, we show the polluted mid-latitude region of Northeast China /
Korea from IAGOS data in 2014-2019, notable for its high ozone values (Gaudel et al., 2020). In
most cases the ozone profiles of Northeast China / Korea are similar to the maximum tropical
ozone profiles, but some regions exceed the Northeast China / Korea ozone values, such as
Southeast Asia / Hanoi, Southern Africa, and the tropical South Atlantic / Ascension Island.
Based on observations from the 1980s and 1990s, ozone levels in the tropics have
generally been considered to be lower than in the mid- and high latitude regions, with the
exception of the tropical Atlantic (Logan et al., 1999; Fishman et al., 1990). However, with
greater availability of ozone profiles across the tropics we can now demonstrate that tropical
India, Southeast Asia, and Malaysia/Indonesia are among the most polluted regions and are
comparable to the mid-latitude regions in terms of ozone pollution (Figure 2). We note that this
unique finding regarding India only pertains to the tropical regions as ozone enhancements
across northern India were detected by the TOMS/SBUV instruments as far back as 1979
(Gaudel et al., 2018).
**3.2 Tropical Tropospheric Column Ozone (TTCO)**






**Figure 3.** Annual tropical tropospheric column ozone (TTCO, surface-270 hPa) from in situ data (IAGOS, SHADOZ between 2014 and 2019 and ATom between 2016 and 2018) (top panel), TTCO (surface-150 hPa) from SHADOZ (2nd panel) between 2014 and 2019; and from OMI (surface to 100 hPa, 2014-2019), OMPS/MERRA2 (surface to potential temperature at 380 K, 2014-2019), TROPOMI (surface to 270 hPa, 2019), CrIS (surface to 2016-2019), OMI/MLS (surface to thermal tropopause, 2014-2019) and IASI/GOME2 (surface to 12 km, 2017-2019).







Figure 3 shows the tropical tropospheric column ozone (TTCO) for SHADOZ, IAGOS
and ATom and for the six-satellite records (OMI, OMPS/MERRA2, TROPOMI, CrIS,
OMI/MLS and IASI/GOME2). As mentioned in Section 2, we focus on the 2014-2019 time
period to study the TTCO distribution. However, ATom data are only available between 2016
and 2018, and some satellite records only cover one or two years within the five-year period we
have chosen. The in situ columns in Dobson units (DU) shown on the first panel of Figure 3 are
from the surface to 270 hPa, with ozone varying between 11 and 33 DU. When the TTCO is
calculated with profiles extending up to 150 hPa (2$^{nd}$ panel of Figure 3 with SHADOZ only),
ozone varies between 18 and 39 DU. As seen with the profiles (section 3.1), the minimum TTCO
values are observed over the Pacific Ocean and the maximum TTCO values are observed over
the Atlantic, Africa, India and Hanoi. The six-satellite records reproduce quite well the
variability of ozone with longitude. However, the range of TTCO values varies by product.
TTCO values under 20 DU are found over the Pacific Ocean with OMI CCD, TROPOMI and
IASI/GOME2, and over Southern Asia with IASI/GOME2. TTCO values above 30 DU are
found over the Atlantic Ocean with all satellite records except IASI/GOME2, and over Africa,
India and Southeast Asia with OMPS/MERRA2, CrIS and OMI/MLS.
Qualitatively, the mid- to upper tropospheric ozone maximum above the Atlantic and
Africa is well known (Fishman et al., 1987; Thompson et al., 2003) and explained by subsidence
of air masses rich in ozone (Krishnamurti et al., 1996; Thompson et al., 2000, 2003), emissions
of lightning NO$_x$ (LiNO$_x$, Sauvage et al., 2007), emissions of CO/VOCs from biomass burning
(Ziemke et al., 2009; Bourgeois et al., 2021) and urban emissions (Tsivlidou et al., 2022). Hanoi,
at the northern edge of our domain, shows previously documented large ozone enhancements
(Ogino et al., 2022), equivalent to those above Africa and the Atlantic. A new maximum,
equivalent to that found above Africa, is now detected over India, mostly related to human
activities (fossil fuel combustion and agriculture burning) (Singh et al., 2020).
However, the accurate quantification of TTCO remains a challenge. The following
section quantifies the differences between the satellite and in situ data in order to improve the
accuracy of TTCO estimates from space.
**3.3 How do the current tropospheric ozone satellite records perform?**

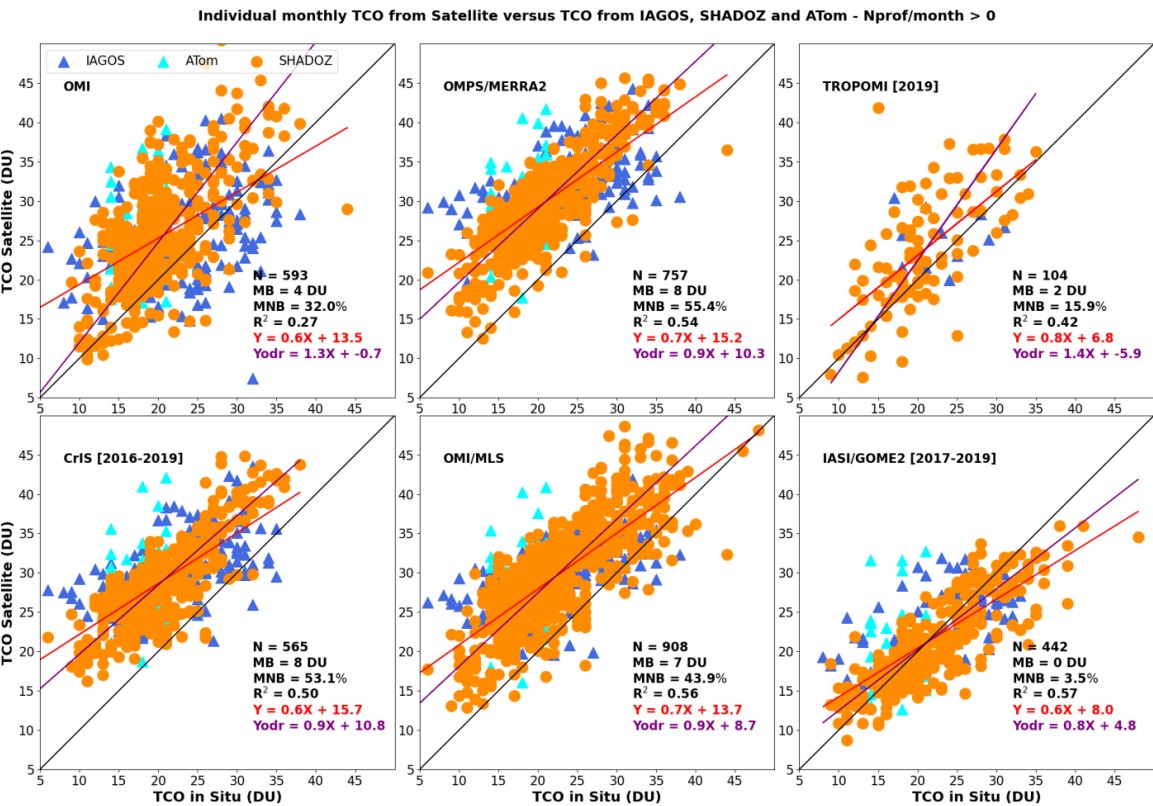

**Figure 4.** Scatter plot of the monthly TTCO from OMI, OMPS/MERRA2, TROPOMI, CrIS, OMI/MLS and IASI/GOME2 satellite records compared with the in situ TTCO from IAGOS (dark blue triangles), ATom (cyan triangles) and SHADOZ (orange circles) between 2014 and 2019. The in situ TTCO values are calculated between the surface and 270 hPa. The TTCO for all satellite data extends much higher (typically up to 100-150 hPa), except for TROPOMI (TTCO calculated from the surface up to 270 hPa) and IASI/GOME2 (TTCO up to 12 km/200 hPa) (Figure S1). The linear least-squares regression is shown in red. The linear orthogonal distance regression is indicated in purple. The number of points (N), the mean biases (MB), the mean normalized biases (MNB) and the correlation coefficient ($R^2$) are shown in black. N corresponds to the number of months with both in situ and satellite data multiplied by the number of IAGOS regions, ATom regions and SHADOZ sites over the time period 2014-2019.

The overall satellite biases of TTCO against in situ TTCO from IAGOS, ATom and SHADOZ are shown in Figure 4. All satellite TTCO values tend to bias high, with mean differences varying from 0 DU to 9 DU. The positive bias is expected since the top level of the satellite TTCO lies higher than that of the in situ data, except for TROPOMI and IASI/GOME2. Figure 4 shows a mean TTCO bias of 2 DU for TROPOMI and no TTCO bias for IASI/GOME2.





For TROPOMI and IASI/GOME2, showing the lowest TTCO biases, the sign of the differences
can change with location (Figure S13). TROPOMI shows positive TTCO biases of 1-4 DU from
the Pacific to Africa and negative biases of 1-2 DU above India, Indonesia/Malaysia.
IASI/GOME2 also shows negative TTCO biases of 1-5 DU above India and Indonesia/Malaysia.
When using only SHADOZ data, rather than all three in situ data sets, as a reference for the
TTCO from the surface to 270 hPa (Figure S14), the mean biases remain the same (compared to
Figure 4), whereas the correlation coefficient and the mean normalized biases increase.

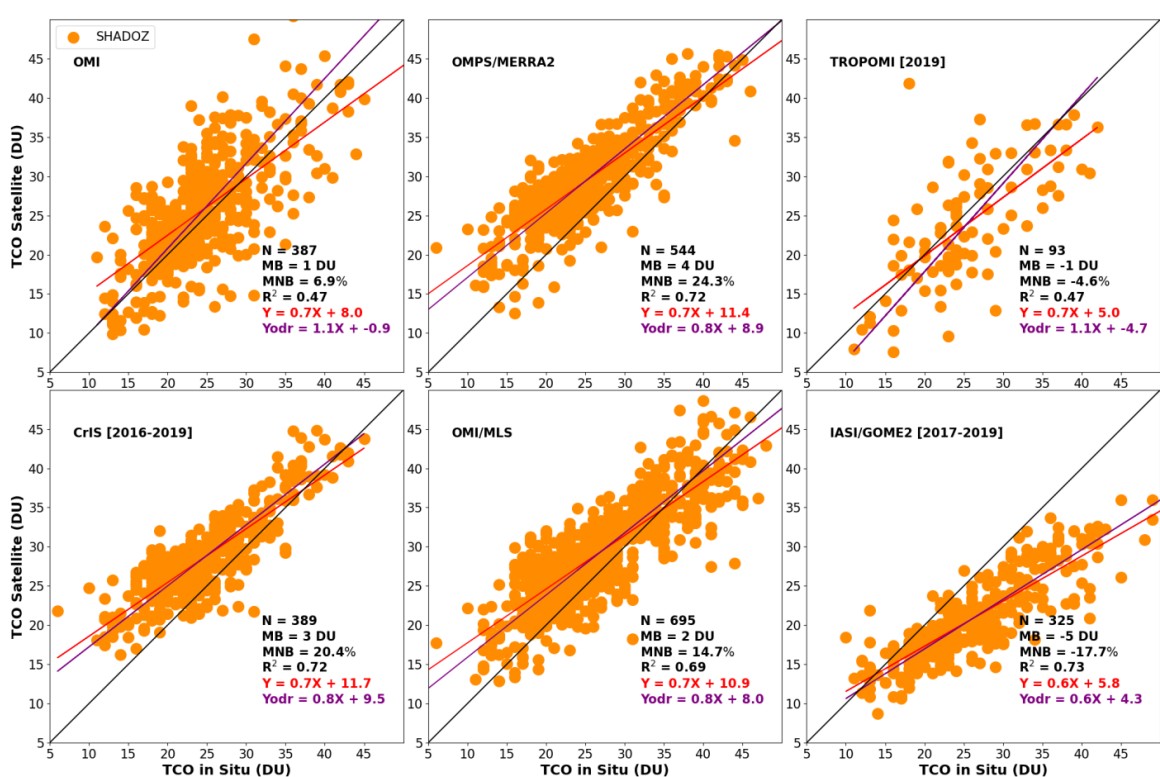

**Figure 5.** Same as Figure 4 but for satellite data compared with SHADOZ TTCO integrated
between the surface and 150 hPa.
Because four satellite records (OMI, OMPS/MERRA2, CrIS and OMI/MLS) show
TTCO from the surface to 100-150 hPa, altitudes that the IAGOS aircraft do not reach, we
compare them to SHADOZ TTCO from the surface to 150 hPa (Figure 5). Both the biases and
the correlation coefficients improve when compared to results for TTCO up to 270 hPa, except
for IASI/GOME2 for which the bias became negative (-5 DU). These results illustrate that
differences in the definition of the top level of the tropospheric column play an important role in





observed differences between satellite TTCO and in-situ TTCO ozone data. There is hence a
need for a common tropospheric column definition to make satellite TTCO estimates comparable
between each other and with in-situ data.
Looking at the SHADOZ sites individually (Figure S15), the biases became closer to zero
above Ascension Island (tropical Atlantic) and Natal (Brazil) when the top level of the column
was changed from 270 hPa to 150 hPa. However, the satellite TTCO records with the top level of
the column higher than 270 hPa (all satellites except TROPOMI and IASI/GOME2) still
overestimate TTCO after changing the reference SHADOZ TTCO's top level from 270 hPa to
150 hPa.
The biases of TROPOMI reported in Figures 4, S11 and 5 are in the range of those
reported in Hubert et al. (2021) with a bias of 2.3±1.9 DU when compared with the SHADOZ
ozonesondes.  Biases estimated for TROPOMI and IASI/GOME 2 using the three in situ TTCO
data sets from the surface to 270 hPa (Figure 4), and biases estimated for OMI,
OMPS/MERRA2, CrIS and OMI/MLS using SHADOZ TTCO from the surface to 150 hPa
(Figure 5) are applied to improve the accuracy of estimates of the tropospheric ozone burden
(TOB), as described in section 3.5.
**3.4 Ozone changes with time**

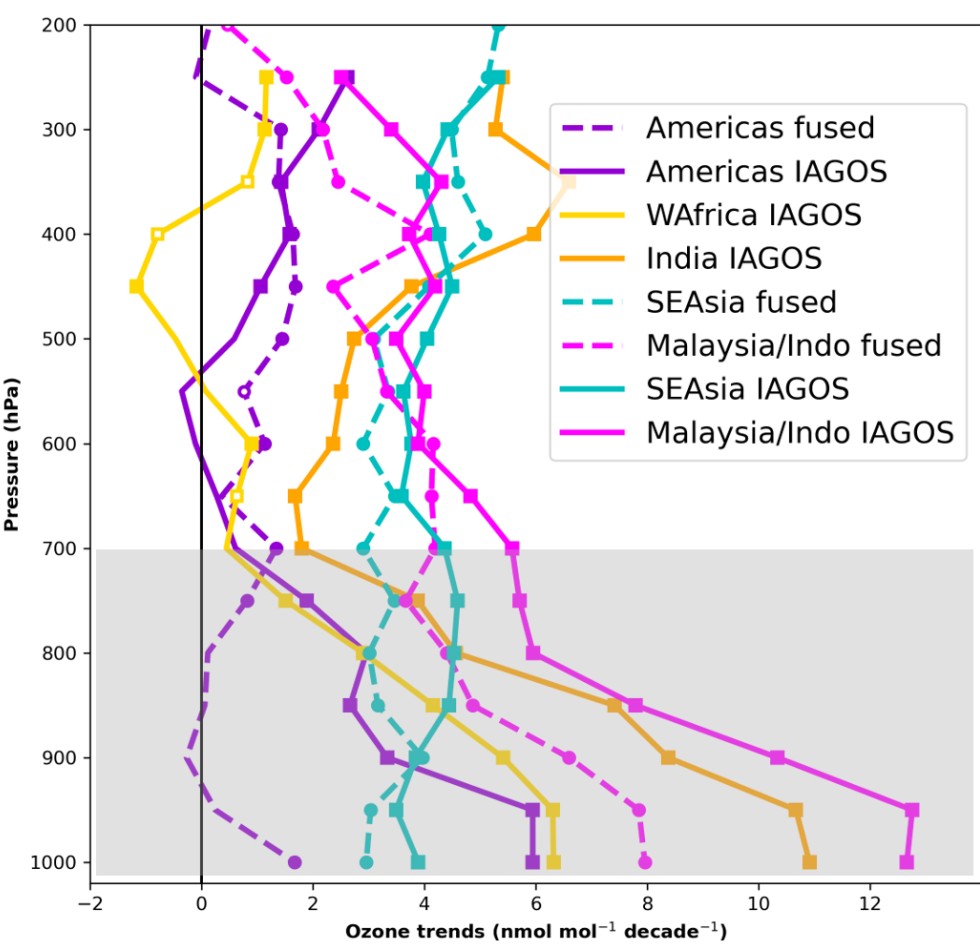

**Figure 6.** Vertical profiles of ozone trends (nmol mol⁻¹ decade⁻¹) between 1994 and 2019, at 50 hPa vertical resolution. Trends are calculated for the 5 IAGOS regions in the tropics: Americas, Western Africa, India, Southeast Asia and Malaysia/Indonesia. SHADOZ data are available for 3 out of the 5 IAGOS regions and used to produce fused trends (IAGOS + SHADOZ). Filled squares (IAGOS trends) or circles (fused trends) indicate trends with *p*-values less than 0.05. Open squares or circles indicate trends with *p*-values between 0.05 and 0.1. The zero-trend value is indicated with a vertical black line. The vertical range below 700 hPa is shaded grey to indicate that the fused trends are based on several sites and airports influenced by different local air masses. The 2-sigma values associated with the ozone trends are shown in Figure S16.





The estimation of trends of tropospheric ozone in the tropics based on in situ observations
is a difficult task as the data are sparse in time and space, as discussed below. In this study, the
Americas, Africa, Southeast Asia and Malaysia/Indonesia are regions sampled both by IAGOS
and SHADOZ allowing us to improve the trends estimate in the free troposphere (above 700
hPa) by fusing both datasets to achieve a greater sample size and a better representation of
regional ozone variability (sections 2.5, S1 and Figures S2-S8). Figure 6 shows trends from the
fused datasets and also from the IAGOS data only. We observed increasing ozone levels between
1994 and 2019 over Americas (trends ranging from $-0.3 \pm 0.6$ to $1.8 \pm 0.7$ nmol mol$^{-1}$ decade$^{-1}$
with the vertical levels), Africa (from $-0.3 \pm 0.6$ to $7.4 \pm 0.4$ nmol mol$^{-1}$ decade$^{-1}$), India (from
$0.9 \pm 1.4$ to $11 \pm 2.4$ nmol mol$^{-1}$ decade$^{-1}$), Southeast Asia (from $2.5 \pm 0.4$ to $5.1 \pm 0.8$ nmol mol$^{-1}$
decade$^{-1}$) and Malaysia/Indonesia (from $0.5 \pm 0.6$ to $8.0 \pm 0.8$ nmol mol$^{-1}$ decade$^{-1}$). In the
boundary layer (<700 hPa), local air masses sampled above SHADOZ sites and IAGOS airports
are likely very different in terms of emissions, photochemistry and airmass history, which may
explain higher differences between the fused and IAGOS trends than in the free troposphere. The
strongest trend we find is $12.5 \pm 2.2$ nmol mol$^{-1}$ decade$^{-1}$ in the boundary layer over
Malaysia/Indonesia using IAGOS data only. Malaysia/Indonesia is the region for which the
number of years with missing data is excessive. However, we do not expect this gap to alter the
trend's estimate because, as mentioned in the Methods section and based on Blot et al. (2020),
MOZAIC and IAGOS data sets are consistent, and together they yield continuous multi-decadal
data records. As shown by Gaudel at al. (2020), the "L" shape of the trends, with a rather
constant trend above the 700 hPa level and larger trends in the boundary layer, is common to the
studied tropical regions except for Southeast Asia, which shows similar trends in both the
boundary layer and in the free troposphere. Taking the fused trends as the reference, we find that
the trends estimated using IAGOS data only tend to be overestimated by 1-2 nmol mol$^{-1}$ decade$^{-1}$
at 700-500 hPa, except over the Americas, and underestimated by 0.5-1 nmol mol$^{-1}$ decade$^{-1}$ at
500-250 hPa, except over Malaysia/Indonesia. Only IAGOS ozone profiles are available over
India and the trends in this region can reach up to $6.7 \pm 1.8$ nmol mol$^{-1}$ decade$^{-1}$ at 350 hPa,
which exceed the trends over the other regions at the same vertical level.



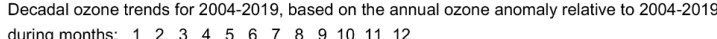

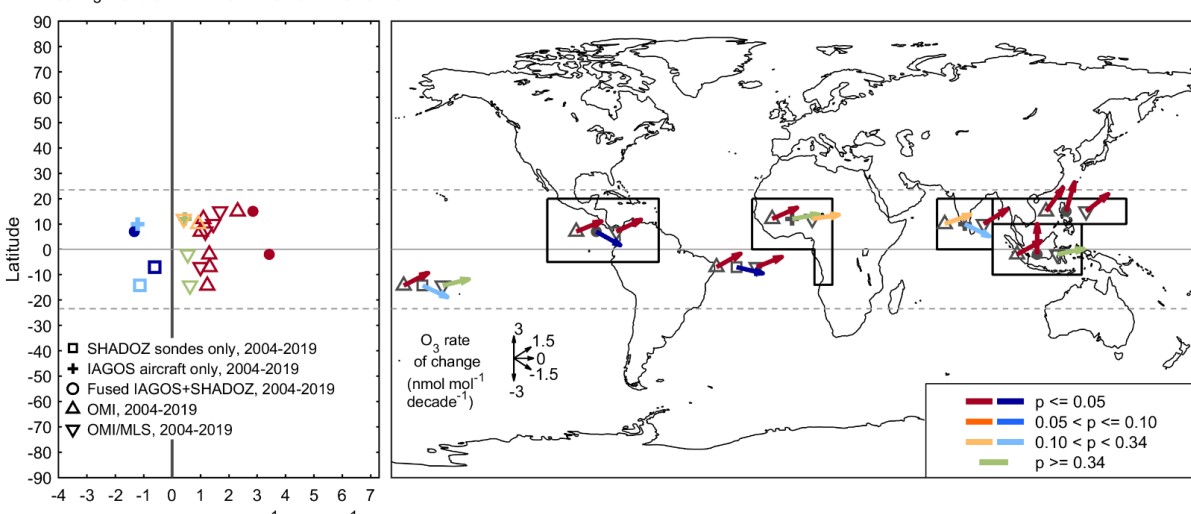

**Figure 7**. TTCO trends (nmol mol$^{-1}$ decade$^{-1}$) between 2004 and 2019 from IAGOS (crosses), SHADOZ (squares), IAGOS fused with SHADOZ (circles), OMI (triangles up) and OMI/MLS (triangles down) above the five continental IAGOS regions (Americas, Africa, India, Southeast Asia and Malaysia/Indonesia) and two oceanic SHADOZ regions (Samoa and Natal + Ascension Island). The left panel shows the trends of ozone as a function of latitude. The right panel shows the trends of ozone on the map with the black rectangles demarcating the five IAGOS regions. On the map, the longitude of the crosses, circles, triangles and squares are arbitrary and the latitude is the mean latitude of the black rectangles or relative to the SHADOZ sites. The direction of the arrows shows the magnitude of the trends and the colors indicate the *p*-value. The TTCO trends from in situ data are calculated from the monthly TTCO between the surface and 100 hPa, except over India where IAGOS profiles are available between the surface and around 200 hPa. The TTCO trends from OMI and OMI/MLS are calculated from the monthly TTCO defined between the surface and around 102-105 hPa (Figure S1).

Satellite data from OMI are available continuously since 2004 and 15-year trends cannow be estimated. The interannual variability of the TTCO from satellite and in situ data is shown in Figure S17. Several time series of the monthly mean of tropospheric ozone above Malaysia/Indonesia show the influence of climate variability such as El Niño and related fires. For example, we see a peak of ozone in September 2015 in agreement with a peak of CO emissions due to biomass burning above Equatorial Asia (Figure S18, Mead et al., 2018). Figure 7 and Table 1 show the trend estimates of TTCO in nmol mol$^{-1}$ decade$^{-1}$ from OMI CCD, OMI/MLS, and in situ data between 2004 and 2019. The in situ trends between 2004 and 2019 (Figure S19 and Table 1) are negative for Samoa (-1.1 ± 1.9 nmol mol$^{-1}$ decade$^{-1}$), Americas (-1.3 ± 0.4 nmol mol$^{-1}$ decade$^{-1}$), Natal/Ascension Island (-0.6 ± 0.5 nmol mol$^{-1}$



decade$^{-1}$) and India (-1.2 ± 1.8 nmol mol$^{-1}$ decade$^{-1}$), and positive for Western Africa (0.4 ± 1
nmol mol$^{-1}$ decade$^{-1}$), Southeast Asia (2.9 ± 1.4 nmol mol$^{-1}$ decade$^{-1}$) and Malaysia/Indonesia
(3.4 ± 1.3 nmol mol$^{-1}$ decade$^{-1}$). The presence of negative trends above some regions for the
shorter 2004-2019 period differs greatly from the longer 1994-2019 time period which had no
time series with negative trends except above Samoa (Figure 7, Table 1). They also differ from
the positive trends shown by the satellite data (full record, Figure 7, Table 1). The satellite trends
vary between 0.9 ± 1.3 nmol mol$^{-1}$ decade$^{-1}$ over India and 2.3 ± 1.3 nmol mol$^{-1}$ decade$^{-1}$ over
Southeast Asia with OMI, and between 0.4 ± 0.8 nmol mol$^{-1}$ decade$^{-1}$ over Western Africa and
1.7 ± 0.8 nmol mol$^{-1}$ decade$^{-1}$ over Southeast Asia with OMI/MLS (Figure 7, Table 1).
Discrepancies between satellites and in situ observations in assessing trends may be
caused by (i) the different definitions of the tropospheric column (100 hPa, 200 hPa or
tropopause defined with the temperature lapse rate); (ii) the diminished sensitivity of the space-
based instruments in the boundary layer; or (iii) the limited data availability and relatively short
record that may lead to less accurate and precise trends (Figures S2 and S3). In particular we
highlight previous research that has demonstrated the difficulty in detecting ozone trends in time
series that are noisy and or sparsely sampled (Weatherhead et al., 1998; Fischer et al., 2011;
Barnes and Fiore, 2016; Fiore et al., 2022). These studies show that 20 years of observations, or
more, are needed for trend detection, and that model ensembles (based on differing initial
conditions) can produce trends for a given location that vary so widely that even the sign can
fluctuate between positive and negative, when dealing with time periods less than 20 years.
Furthermore, previous studies of in situ ozone profiles concluded that a sampling frequency of
once per week generally fails to produce accurate monthly mean and trend values (Logan, 1999;
Saunois et al., 2012; Chang et al., 2020, 2022). Consistent with these previous studies, we
conducted our own analysis of tropical ozone time series (see the Supplementary Section S1) and
found that these sparsely sampled data sets have very low signal-to-noise ratios, which makes
trend detection very difficult, especially when a time series is less than 20 years in length (Chang
et al., 2020,2022). The comparison between the in situ and satellite trends is only 15 years in
length (2004-2019), and the in situ datasets are sparsely sampled, characteristics consistent with
known challenges for trend detection. Furthermore, we point out that the robustness of the
positive trends from the satellite records greatly diminishes, and even becomes undetectable,
when we reduce the sample size of the satellite data in the IAGOS regions to match the sparse
sampling frequency of the aircraft observations (Figure S20). For example, when the satellite
data are fully sampled across the five IAGOS domains, all trends are positive, within the range
0.4 ± 0.8 to 2.3 ± 1.3 nmol mol$^{-1}$ decade$^{-1}$. But when the satellite sample sizes are reduced so
that they only coincide with the specific months and grid-cells sampled by the IAGOS aircraft,
the range of the trends more than doubles and even includes negative values (-3.1 ± 2.6 to +3.6 ±
2.1 nmol mol$^{-1}$ decade$^{-1}$). This increased uncertainty is an expected outcome of decreased
sampling frequency, as illustrated in Figure S2.
The color scheme in Table 1 reflects our overall confidence in the presence of in situ
trend estimates, according to the number of missing monthly values, monthly average data
availability, the length of study period, and the $p$-value of the trend estimate (the trends are
confident only if a low $p$-value and a high data coverage are met, see Appendix A for further
details and Section S3 for a discussion of the confidence assigned to each region). When
assigning a level of confidence to a trend we weigh the p-value and the data coverage and ask the
question: "Are we confident that a positive or negative trend is reliable?" For example, if a
positive trend has a low p-value but also low data coverage then our confidence that the trend is





reliable is diminished.  Western Africa is the only region in this study with sufficient sampling
for reliable trend detection with high confidence (1994-2019).  Trends derived from the other in
situ time series only have low or medium confidence due to sampling deficiencies and/or low
estimation certainty (based on the p-value). When we compare the satellite trends to the in situ
trends we find that they are consistent for Southeast Asia, with all three data sets showing
positive trends. In the other regions we find discrepancies between the in situ and satellite trends,
but in these regions, we do not have high confidence in the in situ trends, and therefore there is
no reason to reject the satellite trend values, which generally indicate an increase of ozone in the
study regions.  However, the discrepancies between satellite and in situ trends in the Americas
and Natal + Ascension Island are nuanced and require further discussion.  In the Americas region
we assigned medium confidence to the decreasing ozone trends based on the in situ observations,
which contrasts strongly with the clear positive trends based on the satellite data.  When we
reduced the satellite sampling coverage to match the locations and months with IAGOS
observations, we found that the satellite trends switched from clear positive trends to clear
negative trends (Figure S20). This exercise indicates that the available in situ observations are
not representative of the large region, and therefore they do not provide sufficient justification
for rejecting the positive trends reported by the satellite data. In situ ozone trends above Natal +
Ascension Island have a weak negative trend ($-0.62\pm0.54$ nmol mol$^{-1}$ decade$^{-1}$) with medium
confidence, while the satellite trends show weak positive trends.  While the divergence between
the positive and negative trends is small over this short time period (2-3 nmol mol-1 over 15
years), this discrepancy warrants further investigation to determine the differences between the
satellite and in situ time series trends.

## 3.5 Comparison to previous studies

Using the ozonesondes from the SHADOZ network, Thompson et al. (2021) found
positive annual trends of about $1.2 \pm 3$ % decade$^{-1}$ to $1.9 \pm 3$ % decade$^{-1}$ ($0.08 \pm 1.68$ nmol mol$^{-1}$
decade$^{-1}$ to $0.78 \pm 1.66$ nmol mol$^{-1}$ decade$^{-1}$) between 1998 and 2019 at 5-10 km (~500-250 hPa)
across the tropical belt. They reported maximum trends ($1.9 \pm 3$ % decade$^{-1}$) above the
Malaysia/Indonesia (Kuala Lumpur + Java) and Americas (San Cristobal + Paramaribo) regions
and minimum trends ($1.2 \pm 3$ % decade$^{-1}$) above Africa (Nairobi). The SHADOZ trends are
slightly lower than the IAGOS + SHADOZ fused trends or IAGOS trends which may be
explained by the different starting points of the time series (1998 for SHADOZ data and 1994 for
IAGOS data), but they are all positive.
Previous studies of TTCO trends from satellite data relied on data harmonization in order
to combine several satellite records into a time series spanning at least two decades and to better
account for the climate variability in the trend estimates (Heue et al., 2016; Leventidou et al.,
2018; Ziemke et al., 2019; Pope et al., 2023).  Heue et al. (2016) found a tropical trend of $0.7 \pm$
$0.12$ DU decade$^{-1}$, with regional trends ranging from $+1.8$ DU decade$^{-1}$ on the African Atlantic
coast, to $-0.8$ DU decade$^{-1}$ over the western Pacific Ocean.  Leventidou et al. (2018) reported
positive trends of TTCO of 1 to 1.5 DU decade$^{-1}$ between 1996 and 2015 over Northern South
America, North Africa, South Africa and India, and negative trends of $-1.2$ to $-1.9$ DU decade$^{-1}$
above the oceans (Pacific, Atlantic, Indian oceans). Using TOMS-OMI/MLS, Ziemke et al.
(2019) reported positive trends between 1979 and 2016 across the tropical latitude band 20˚S-
20˚N except above the southeastern tropical Pacific Ocean and southeastern Indian Ocean. The
highest positive trends (up to 1.3 DU decade$^{-1}$) were found above South-Southeast Asia and
Central Africa. Finally, a new harmonized product that quantifies ozone between the surface and



450 hPa reports much higher tropical trends than the other studies, with increases of 2.9 ± 1.6
DU decade$^{-1}$ for the southern tropical band (0 – 15º S) and 3.9 ± 1.8 DU decade$^{-1}$ for the northern
tropical band (0 – 15º N) for the years 1996-2017 (Pope et al., 2023).  While these findings vary
regarding the magnitude of trends in the tropics, when taken into consideration with the 1994-
2019 in situ trends reported by the present study, the preponderance of evidence indicates a
general increase of TTCO since the mid-1990s.
Wang et al. (2022) report an increase of TTCO (950 - 250 hPa) trends using the GEOS-
Chem chemical transport model above the IAGOS' regions and SHADOZ sites between 1995
and 2017, except above Samoa. The trends vary with locations between -0.60 ± 0.38 nmol mol$^{-1}$
decade$^{-1}$ above Samoa and 2.87 ± 0.23 nmol mol$^{-1}$ decade$^{-1}$. In general, they find that the TTCO
trends from the model are lower by 1-3 nmol mol$^{-1}$ decade$^{-1}$ than from the observations, except
above Paramaribo.





**Table 1. Summary of the TTCO trends in nmol mol$^{-1}$ decade$^{-1}$ from IAGOS, SHADOZ,**
**OMI/MLS and OMI CCD.**
The sampling column reports three numbers for the in situ data: i) the number on the top refers to
the average number of profiles per months taking into account all the months with profiles, ii)
the number in the middle refers to the percentage of months with data for the studied time-period
(1994-2019 or 2004-2019), iii) the number in the bottom refers to the total number of profiles for
the studied time period (1994-2019 or 2004-2019). For the satellites, the sampling column
reports "Full" when the full record is taken into account and "Filtered" when the satellite sample
sizes have been greatly reduced so that they only coincide with the specific months and grid-cells
sampled by the IAGOS aircraft. The table cells are color coded to reflect the low confidence
(light blue), medium confidence (blue) and high confidence (dark blue) on the ozone trends from
the in situ data based on the sampling and the p-value.

| | | 1994-2019 | | | 2004-2019 | | |
|---|---|---|---|---|---|---|---|
| | | Trends±2σ (nmol mol$^{-1}$ decade$^{-1}$) | p-value | Sampling | Trends±2σ (nmol mol$^{-1}$ decade$^{-1}$) | p-value | Sampling |
| IAGOS | Western Africa | 2.34±0.48 | <0.01 | 18.8 71.8% 3411 | 0.44±1.04 | 0.40 | 20.2 66.7% 2261 |
| | India | 5.68±1.06 | <0.01 | 7.6 66.7% 1574 | -1.21±1.76 | 0.17 | 8.5 67.7% 1100 |
| SHADOZ | Samoa | -0.03±1.21 | 0.97 | 3.2 92.8% 779 | -1.13±1.90 | 0.23 | 3.1 91.6% 537 |
| | Natal + Ascension Island | 0.49±0.49 | 0.04 | 6.3 90.4% 1426 | -0.62±0.54 | 0.01 | 6.0 87.2% 939 |
| Fused IAGOS + SHADOZ | Americas | 0.47±0.79 | 0.36 | 12.2 92.2% 3642 | -1.33±0.39 | <0.01 | 10.7 93.6% 2036 |
| | Southeast Asia | 3.51±0.78 | <0.01 | 11.2 77.8% 2501 | 2.85±1.38 | <0.01 | 10.2 82.8% 1730 |
| | Malaysia/Indonesia | 3.96±0.53 | <0.01 | 5.0 89.8% 1445 | 3.42±1.35 | <0.01 | 4.7 89.9% 954 |
| OMI CCD | Americas | | | | 1.01±0.72 | 0.01 | Full |
| | | | | | -3.06±2.65 | 0.02 | Filtered |
| | Western Africa | | | | 1.10±1.04 | 0.04 | Full |
| | | | | | -1.04±3.08 | 0.50 | Filtered |
| | India | | | | 0.92±1.26 | 0.15 | Full |
| | | | | | 1.20±2.95 | 0.42 | Filtered |
| | Southeast Asia | | | | 2.31±1.34 | <0.01 | Full |
| | | | | | 3.56±2.08 | <0.01 | Filtered |
| | Malaysia/Indonesia | | | | 1.31±1.15 | 0.02 | Full |
| | | | | | 2.26±3.42 | 0.19 | Filtered |
| | Samoa | | | | 1.24±1.17 | 0.04 | |
| | Natal + Ascension Island | | | | 1.32±1.04 | 0.01 | |
| OMI/MLS | Americas | | | | 1.17±0.72 | <0.01 | Full |





| | | | | | | | |
|---|---|---|---|---|---|---|---|
| | | | | | -2.79±1.96 | 0.01 | Filtered |
| | Western Africa | | | | 0.41±0.80 | 0.30 | Full |
| | | | | | 0.68±3.95 | 0.73 | Filtered |
| | India | | | | 1.45±0.79 | <0.01 | Full |
| | | | | | -1.64±1.67 | 0.05 | Filtered |
| | Southeast Asia | | | | 1.69±0.83 | <0.01 | Full |
| | | | | | 2.46±1.85 | 0.01 | Filtered |
| | Malaysia/Indonesia | | | | 0.55±1.22 | 0.37 | Full |
| | | | | | 1.39±4.36 | 0.53 | Filtered |
| | Samoa | | | | 0.63±1.34 | 0.35 | |
| | Natal + Ascension Island | | | | 1.00±0.78 | 0.01 | |





**3.6 Tropical tropospheric ozone burden**

**Figure 8.** Time series of tropospheric ozone burden (Tg) from OMI/MLS, OMPS/MERR2, OMI
CCD, TROPOMI, CrIS and IASI/GOME2. The panels show the monthly means for the Northern
Hemisphere (a and b) and the Southern Hemisphere (c and d) before and after bias correction.
The biases we used are in DU and from the differences between IASI/GOME2 and TROPOMI
TTCO using the reference TTCO up to 270 hPa and between OMI, OMI/MLS,
OMPS/MERRA2, CrIS TTCO using the reference TTCO up to 150 hPa (Figures 4 and 5).


Figure 8 shows the time series of the tropical tropospheric ozone burden (TTOB, Tg)
from six satellite records. As described in the Methods, OMI/MLS, OMPS/MERRA2,
TROPOMI, CrIS, and IASI/GOME2 are sampled in the 20˚S-20˚N latitude band, while OMI is
constrained to the 15˚S-15˚N latitude band. For both hemispheres we find two distinguished
groups in terms of TTOB (Figure 8, panels a and c): (i) OMI CCD, TROPOMI and
IASI/GOME2 with a range of TTOB of 25-45 Tg, (ii) OMI/MLS, OMPS/MERRA2 and CrIS
with a range of TTOB of 40-65 Tg. These differences are explained by the difference of latitude
coverage (OMI CCD) and the upper bound of the tropospheric column (lower for TROPOMI and



IASI/GOME2 than for the other satellite data). The seasonal variability of TTOB is lower in the
northern hemisphere than in the southern hemisphere especially in the narrowest latitude band
(OMI CCD).
The biases calculated from the scatter plots of satellite versus ozonesondes (Figures 4 and
5) are used to correct the satellite time series. The adjustment reduced the differences by about
10 Tg in the northern hemisphere and by 5 Tg in the southern hemisphere, between the two
groups mentioned above. In the northern hemisphere, after adjustment (Figure 8, panels b and d,
and Table 2), IASI/GOME TTOB (45-55 Tg) become part of group ii) and TROPOMI TTOB
(30-43 Tg) moved closer to OMI CCD TTOB in group i). In the southern hemisphere, it is
difficult to distinguish the two groups, after adjustment. For example, the range of
OMPS/MERRA2 TTOB (30-60 Tg) covers the range of the TTOBs in both groups after 2019.
Table 2 summarizes TTOB trends from this study, and from TOAR-Climate (Gaudel et
al., 2018). Trends are positive and higher in the northern hemisphere ($1.4 \pm 0.7$ Tg decade$^{-1}$ to
$5.7 \pm 2.5$ Tg decade$^{-1}$) than the southern hemisphere ($0.9 \pm 2.2$ Tg decade$^{-1}$ to $5.1 \pm 4.5$ Tg
decade$^{-1}$). Because TTOB trends in Tg decade$^{-1}$ can increase with the width of the latitude band
(assuming trends are all positive across the range of latitudes considered), we also report trends
in % decade$^{-1}$, to compare trends between different latitude bands. We found that trends in the 0-
15° (OMI CCD) north and south latitude bands are lower than in the 0-20° (OMI/MLS) latitude
bands by 2-4 % decade$^{-1}$. These differences might be explained by a quicker increase of
tropospheric ozone in the subtropics than the equatorial region, or by a potential discrepancy
between OMI CCD and OMI/MLS. It is worth noting that the 2004-2016 OMI/MLS trends in the
0-30° north and south latitude bands are higher by a factor of 3 or 5 than the 2004-2019
OMI/MLS trends in the 0-20° north and south latitude bands. These differences might also be
explained by the influence of the larger increases of subtropical tropospheric ozone.



**Table 2**. Summary of tropical tropospheric ozone burden values and trends.

| | Latitude band | Tropospheric Ozone Burden | | | Trends | | | |
|---|---|---|---|---|---|---|---|---|
| | | Period | Instrument/ model | Values Tg | Period | Instrument | Values Tg/decade | Values %/decade |
| This study (These numbers are corrected using bias results from Figure 5) | 0-15˚N | 2004-2019 | OMI | **31.8** ± 4.3 | 2004-2019 | OMI | **1.4** ± 0.7 | **4** ± 2 |
| | 0-15˚S | 2004-2019 | OMI | **32.9** ± 11.7 | 2004-2019 | OMI | **0.9** ± 2.0 | **2** ± 6 |
| | 0-20˚N | 2004-2021 2012-2021 2016-2021 2017-2021 2019 | OMI/MLS OMPS CrIS IASI/GOME 2 TROPOMI | **46.6** ± 7.0 **48.1** ± 7.4 **46.4** ± 7.5 **38.1** ± 5.9 **34.9** ± 5.1 | 2004-2019 | OMI/MLS | **1.6** ± 1.1 | **3** ± 2 |
| | 0-20˚S | 2004-2021 2012-2021 2016-2021 2017-2021 2019 | OMI/MLS OMPS CrIS IASI/GOME 2 TROPOMI | **44.9** ± 13.0 **45.3** ± 15.1 **44.6** ± 13.4 **37.1** ± 8.6 **34.7** ± 10.7 | 2004-2019 | OMI/MLS | **0.9** ± 2.2 | **2** ± 5 |
| TOAR-Climate (Figures S28, S29) | 0-30˚N | | | | 2004-2016 | OMI/MLS | **5.7** ± 2.5 | **7** ± 3 |
| | 0-30˚S | | | | 2004-2016 | OMI/MLS | **5.1** ± 4.5 | **6** ± 5.6 |




**4. Conclusions**

Long and mid-term records of tropospheric ozone from IAGOS, SHADOZ, and OMI, as well as new observations from the ATom aircraft campaign and the CrIS, IASI/GOME2 satellite instruments are now available in the tropics, a region undergoing rapid changes in terms of human activity and emissions of ozone precursors. The present study takes advantage of these new data records to assess the distribution of tropical tropospheric ozone, and it uses the longest records to assess its trends:

**Present-day distribution**

- With greater availability of ozone profiles across the tropics we can now demonstrate that southern India is among the most polluted regions (Western Africa, tropical South Atlantic, Southeast Asia, Malaysia/Indonesia) with 95th percentile ozone values reaching 80 nmol mol$^{-1}$ in the lower free troposphere, comparable to mid-latitude regions, such as Northeast China/Korea.
- The lowest ozone values (5th percentile) are less than 10 nmol mol$^{-1}$, and are observed by SHADOZ and ATom in the boundary layer (below 700 hPa) above the Americas and the tropical South Pacific.
- From space, the distribution of tropical tropospheric column ozone (TTCO) varies among the satellite products by 5-10 DU in the 20ºS-20ºN latitude band.
- The satellite data tend to overestimate tropical ozone with mean biases (between the surface and 270 hPa) ranging between 0 for IASI/GOME2 and 9 DU for OMPS/MERRA2 when compared to IAGOS, ATom and SHADOZ.
- The smallest biases ($\leq$ 2 DU) are found when matching the top limit of the in situ profiles to that of the OMI, TROPOMI and IASI/GOME2 satellite records.
- The in situ observations were critical for adjusting the biases in the satellite products, bringing them into closer alignment. The TTOB is about 31.5 Tg in both tropical hemispheres up to 15˚N or 15˚S. The TTOB is larger in the northern hemisphere than in the southern hemisphere by about 2 Tg when considering the larger latitude band between 20˚S and 20˚N. The seasonal variability of TTOB is weaker closer to the equator in the northern hemisphere.

**Trends**

- When focusing on the longest available records exceeding 20 years (1994-2019, IAGOS/SHADOZ data reported in this study) or 30 years (1979-2016 satellite record reported by Ziemke et al., 2019) we see a consistent picture of increasing ozone across the tropics. IAGOS and SHADOZ data were fused to increase the sample sizes and to improve the statistics of the data over three out of the five IAGOS regions: Americas, Southeast Asia, Malaysia/Indonesia (Western Africa and India with no SHADOZ data). India and Malaysia/Indonesia are the regions with the strongest ozone increase below 800 hPa (11 $\pm$ 2.4 and 8 $\pm$ 0.8 nmol mol$^{-1}$ decade$^{-1}$ close to the surface, respectively) and India above 400 hPa (up to 6.8 $\pm$ 1.8 nmol mol$^{-1}$ decade$^{-1}$). Southeast Asia and Malaysia/Indonesia show the highest increase in the mid-troposphere (550-750 hPa, up to 3.4 $\pm$ 0.8 and 4 $\pm$ 0.5 nmol mol$^{-1}$ decade$^{-1}$, respectively). Trends of the tropical tropospheric column ozone reflect these results. In terms of in situ trend reliability based on data availability and *p*-value of trend estimate, we have the most confidence in



Western Africa (while it is still not ideal due to moderate data gaps) and the least
confidence in Samoa and Americas.
●  For shorter time periods (< 20 years) trend detection can be even more challenging due to
the larger additional uncertainty associated with sparsely sampled ozone records.
●  The OMI and OMI/MLS satellite records have a very high sampling frequency compared
to the sparse in situ datasets and mostly show positive 15-year (2004-2019) trends above
the IAGOS regions (from $0.55 \pm 1.22$ to $2.31 \pm 1.34$ nmol mol$^{-1}$ decade$^{-1}$) with the
maximum trends over Southeast Asia of $2.31 \pm 1.34$ nmol mol$^{-1}$ decade$^{-1}$ with OMI CCD,
and $1.69 \pm 0.89$ nmol mol$^{-1}$ decade$^{-1}$ with OMI/MLS. The strongest agreement between
satellite and in situ trends is found above Southeast Asia where TTCO had increased at a
rate of about 2-3 nmol mol$^{-1}$ decade$^{-1}$. These trends are consistent with the results from
Ziemke et al. (2019) using TOMS-OMI/MLS records and Gaudel et al. (2020) using
IAGOS ozone profiles. Above the other regions, we only have low to medium confidence
in the in situ trends, therefore we concluded that we have no reason to reject the positive
tropical tropospheric ozone trends based on satellite data. However, the discrepancy
between the weak positive satellite trends and the weak negative in situ trends above
Natal + Ascension Island warrants further investigation.

This study demonstrates that most tropical regions require either an increased and/or
continuous sampling (in situ and remote sensing) of ozone because either there are no data, or
the data are so sparse that it is difficult to estimate accurate and precise trends to evaluate the
satellite records. However, we also demonstrate that the current sampling frequency is adequate
for bias correcting the satellite products, as shown in Figure 8.
TROPOMI, IASI/GOME2, CrIS and OMPS/MERRA2 are recently available satellite
records and their overlap for several years with the OMI record will assure continuity of ozone
and precursors observations from space when the NASA Aura mission terminates by 2025.
GEMS, the only geostationary mission covering the tropics (tropical Asia), will bring new
capabilities in monitoring the region with the strongest ozone increases in the world, with higher
spatial and temporal resolution than the polar orbiting instruments.
This study underscores the importance of developing TTCO data with a common
definition of the top of the tropospheric column. Additionally, there is a pressing need for the
availability of common or joint retrievals for satellite data, such as those provided by initiatives
like TROPESS (TRopospheric Ozone and its Precursors from Earth System Sounding,
https://tes.jpl.nasa.gov/tropess/).
Moreover, to better understand the drivers behind the observed increases in TTOB, it is
essential to conduct simulations using global chemical transport models, chemistry climate
models, Earth system models, and regional models spanning recent decades. Encouragingly,
these endeavors have been newly proposed within the framework of the Tropospheric Ozone
Assessment Report phase II (TOAR-II), an initiative under the International Global Atmospheric
Chemistry (IGAC) project. These efforts will be the focus of forthcoming publications featured
in the TOAR-II Community Special Issue.



**Appendix A**

The Intergovernmental Panel on Climate Change (IPCC) developed a guidance note for the consistent treatment of uncertainties (Mastrandrea et al., 2010) that was followed by the fifth and sixth IPCC assessment reports. Among other applications, the calibrated language described by the guidance note is helpful for the discussion of long-term trends and for communicating the level of confidence that an author team wishes to assign to a particular trend value, or to an ensemble of trend values. Confidence in the validity of a finding is expressed qualitatively with five qualifiers (very low, low, medium, high and very high), based on the type, amount, quality, and consistency of the available evidence, and the level of agreement among studies addressing the same phenomenon (see Figure 1 of Mastrandrea et al., 2010).

Following IPCC, the Tropospheric Ozone Assessment Report (TOAR) developed its own guidance note on best statistical practices for TOAR analyses, featuring an uncertainty scale for assessing the reliability and likelihood of the estimated trend (Chang et al., 2023). The uncertainty scale has five qualifiers as follows: very low certainty or no evidence, low certainty, medium certainty, high certainty and very high certainty. Each qualifier corresponds to a range of values associated with either the signal-to-noise ratio or the $p$-value of the trend. A limitation of the uncertainty scale is that it is best suited for surface ozone time series with high frequency sampling, which allows for robust calculation of monthly means, upon which the trends are calculated. For the case of calculating trends based on sparse ozone profiles, in many cases the monthly means are biased or unreliable due to low sampling frequency, which adds additional uncertainty to the calculation of the trend. Because the $p$-value (or the signal-to-noise ratio) of a trend based on monthly means does not consider the impact of low sampling frequency on the monthly means, we developed new calibrated language to express our confidence in trends based on sparse ozone profiles.

Following the methodology of IPCC (Mastrandrea et al., 2010) Table A1 presents a confidence scale that we use in this present study to express our confidence in a trend based on sparse ozone profiles (as reported in Table 1 in the main text). Any line fit though a time series will produce a trend value that is either positive or negative, and we use this scale to answer the question: "Are we confident that a positive or negative trend is reliable?". The confidence scale considers both data coverage (based on the number of profiles per month and continuity of sampling) and the estimation of the uncertainty of the trend, based on the $p$-value and the 95% confidence interval. Higher confidence can be placed on trends with lower $p$-values and greater data coverage, while less confidence is placed on trends with relatively high $p$-values and low data coverage. The selection of a particular confidence level is qualitative, with no sharp boundaries, however the following guidelines inform our decision-making:

**Data coverage:** Previous studies (Logan, 1999; Saunois et al. 2012; Chang et al., 2020) have shown that sampling rates of once per week (or less) fail to provide accurate monthly means, while increased sampling rates of 2 or 3 times per week are more accurate. The most accurate sampling rate is 4 times per week or higher. Continuous data records with no, or limited gaps, are more reliable than records with multiple or large gaps. Data length also plays a role in trend



reliability. A time series with more than 90% of months with data, and with more than 15
profiles per month is considered to have high data coverage. A time series with 66 to 90% of
months with data, and with 7-15 profiles per month is considered to have moderate data coverage
(this also applies to a region that only meets one condition for high data coverage). A time series
that has less than 66% of months with data, or less than 7 profiles per month has low data
coverage. It should be noted that, based on our criteria, none of the current study regions meet
the criteria for high data coverage, and therefore the top row in Table A1 is not applicable to this
study. In addition, since we derive the trends based on either a 25-year or a 15-year record, it is
natural to consider the trends derived from a longer data record are more robust, as a record
length less than two decades is generally insufficient to eliminate the impact of interannual
variability (Weatherhead et al., 1998; Barnes et al., 2016; Fiore et al., 2022). Therefore, all of the
time series in Table 1 with 15-year records are considered to have low data coverage.

**Estimation uncertainty:** In general, lower $p$-values and higher signal-to-noise ratios are
indicators of a robust trend. The "Guidance note on best statistical practices for TOAR analyses"
(Chang et al., 2023) assigns the following degrees of certainty according to $p$-value: very high
certainty ($p \leq 0.01$), high certainty ($0.05 \geq p > 0.01$), medium certainty ($0.10 \geq p > 0.05$), low
certainty ($0.33 \geq p > 0.10$), very low certainty or no evidence ($p > 0.33$). We acknowledge that
the trends calculation does not consider the inherent quality of the data (i.e. accuracy and
precision of the data), which will be explore in future studies within TOAR Phase II.

**Table A1**. Calibrated language for discussing confidence in long-term trend estimates based on
ozone profiles. Data coverage refers to the number of ozone profiles in a month, and the number
of months with available data. Estimation uncertainty refers to the uncertainty of a trend line
drawn through monthly means, as quantified by the p-value and the 95% confidence interval.

| ↑ **Data coverage**<br><br>(based on the number of profiles per month and continuity of sampling) | **medium confidence**<br>low estimation certainty<br>high data coverage | **high confidence**<br>moderate estimation certainty<br>high data coverage | **very high confidence**<br>high estimation certainty<br>high data coverage |
|---|---|---|---|
| | **low confidence**<br>low estimation certainty<br>moderate data coverage | **medium confidence**<br>moderate estimation certainty<br>moderate data coverage | **high confidence**<br>high estimation certainty<br>moderate data coverage |
| | **very low confidence or no evidence**<br>low estimation certainty<br>low data coverage | **low confidence**<br>moderate estimation certainty<br>low data coverage | **medium confidence**<br>high estimation certainty<br>low data coverage |
| | **Estimation uncertainty →**<br>(based on p-value) | | |





**Author contributions**

Conception and design of the study: AG, IB, ML, K-LC, OC

Generation, collection, assembly, analysis and/or interpretation of data: AG, IB, ML, K-LC, OC

JZ, BS, AMT, RS, DEK, NS, DH, AK, JC, K-PH, PV, KA, JP, CT, TBR

Drafting and/or revision of the manuscript: AG, IB, ML, K-LC, OC, JP, KA, AMT, RS, DH,

AK, NS, JZ, GJF, BCM

All authors approved for submission of the manuscript.

**Competing interests**

ORC is the Scientific Coordinator of the TOAR-II Community Special Issue, to which this paper

has been submitted, but he is not involved with the anonymous peer-review process of this or

any of the other papers submitted to the Special Issue journals.





**Acknowledgements**

This research was supported by the NOAA Cooperative Agreement with CIRES,
NA17OAR4320101 and NA22OAR4320151.
We acknowledge the support of the NOAA JPSS PGRR program.
The authors acknowledge the strong support of the European Commission, Airbus and the
airlines (Lufthansa, Air France, Austrian, Air Namibia, Cathay Pacific, Iberia and China
Airlines, so far) who have carried the MOZAIC or IAGOS equipment and performed the
maintenance since 1994. In its last 10 years of operation, MOZAIC has been funded by INSU-
CNRS (France), Météo-France, Université Paul Sabatier (Toulouse, France) and the Jülich
Research Center (FZJ, Jülich, Germany). IAGOS (https://www.iagos.org/) has been additionally
funded by the EU projects IAGOS-DS and IAGOSERI. The IAGOS database is supported by
AERIS, the French portal for data and service for the atmosphere (see https: //iagos.aeris-
data.fr, last access: April 2023). SHADOZ data are provided through support of NASA's Upper
Atmospheric Composition (UACO), NOAA/Global Monitoring Division and operators and data
archivists across 20 organizations in North and South America, Europe, Africa and Asia.

**Data Availability**

The monthly quasi-biennial oscillation values can be found at https://www.geo.fu-berlin.de/met/ag/strat/produkte/qbo/qbo.dat.

The monthly El Niño-Southern Oscillation index can be found at https://psl.noaa.gov/enso/mei/.

ATom data are archived at https://daac.ornl.gov/ATOM/guides/ATom_merge.html and are published through the Distributed Active Archive Center for Biogeochemical Dynamics (Wofsy et al., 2018).

IAGOS ozone profiles can be found at https://iagos.aeris-data.fr/.

SHADOZ ozone profiles can be found at https://tropo.gsfc.nasa.gov/shadoz/ (see reference list).

IASI+GOME2 satellite data can be found at https://iasi.aeris-data.fr/o3_iago2/, last access 08/02/2023.

OMI CCD, OMI/MLS and OMPS/MERRA2 can be found at https://acd-ext.gsfc.nasa.gov/Data_services/cloud_slice/.

TROPOMI CCD can be found at NASA EarthData repository: https://disc.gsfc.nasa.gov/datasets?keywords=tropomi&page=1

CrIS can be found at https://disc.gsfc.nasa.gov/ (see the Method section for more details on the data preprocessing)



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
