# Peer review of "Tropical tropospheric ozone distribution and trends from in situ and satellite"

_EGUsphere, 2023_

## Community Comment (CC1)

Comments by Rodrigo J. Seguel on behalf of the TOAR-II Steering Committee on:

**Tropical tropospheric ozone distribution and trends from in situ and satellite data**

Audrey Gaudel (corresponding author), Ilann Bourgeois, Meng Li, Kai-Lan Chang, Jerald Ziemke, Bastien Sauvage, Ryan M. Stauffer, Anne M. Thompson, Debra E. Kollonige, Nadia Smith, Daan Hubert, Arno Keppens, Juan Cuesta, Klaus-Peter Heue, Pepijn Veefkind, Kenneth Aikin, Jeff Peischl, Chelsea R. Thompson, Thomas B. Ryerson, Gregory J. Frost, Brian C. McDonald, Owen R. Cooper

This manuscript was submitted to ACP as part of the TOAR-II Community Special Issue https://doi.org/10.5194/egusphere-2023-3095
Discussion started: 22 January 2024; discussion closes 18 April 2024

This review is by Rodrigo Seguel, member of the TOAR-II Steering Committee. The primary purpose of these reviews is to identify any discrepancies across the TOAR-II submissions, and to allow the author teams time to address the discrepancies. Additional comments may be included with the reviews.

While members of the TOAR Steering Committee may post open comments on papers submitted to the TOAR-II Community Special Issue, they are not involved with the decision to accept or reject a paper for publication, which is entirely handled by the journal's editorial team.

**General comments**

The authors have assessed the distribution and trends of tropical tropospheric ozone using available ozone profiles measured by a suite of in situ instruments (IAGOS commercial aircraft, the SHADOZ network, and the ATom aircraft campaign) and six satellite records of tropical tropospheric column ozone (TROPOMI, OMI, OMI/MLS, OMPS/MERRA2, CrIS, and IASI/GOME2). The authors have performed a great job of enhancing the trend detectability and comparability of different data sources and also provided a thorough discussion about bias between satellite and in situ data.

In particular, given the sparsity of the in-situ sampling over the tropics (time and space), SHADOZ and IAGOS measurements were fused over some regions to enhance the trend detection, which is based on quantile regression, as suggested by TOAR-II guidelines (TOAR-II Recommendations for Statistical Analysis). Also, the ozone profiles from in situ observations were converted to columns to evaluate the satellite products and adjust the satellite biases, thus allowing the reduction of the satellite differences in the tropical tropospheric ozone burden (TTOB).

Overall, the findings are consistent with the papers from TOAR-I and the papers submitted to the TOAR-II Community Special Issue. In this regard, the low ozone levels found over the Americas are consistent with the relatively low ozone mixing ratios measured at ground level in the South American tropics compared with South American extratropics values reported in: https://doi.org/10.5194/egusphere-2024-328

**Minor comments**

In section 2 (Methods), the authors clearly define the latitudinal band corresponding to the tropics for this study, which also follows the TOAR-II recommendation (TOAR-II Community Special Issue Guidelines). However, in lines **124** and **132**, they refer to the

tropics between 30º S and 30º N, which includes the subtropics. For consistency, I suggest changing "tropics" to "tropics and subtropics," similar to line **101**.

Line 215-2016 (Caption of Figure 1): Please change "Africa, South Asia" to "Western Africa, India"

Line 257: Do you mean the disagreement is within ± 2%?

Line 283: Please check the parenthesis.

Line 699: Change "cannow" to "can now"

Line 894-896 (Conclusions): In addition to the processes described by Kley et al. (1996), are there any other relevant processes that explain the low ozone values found in the Americas and the tropical South Pacific that may be suitable to discuss?

---

## Author Comment (AC1)

**Response to the Reviewers' comments**

**Review 1**

This manuscript examines tropical tropospheric ozone and its trends across different observational datasets including in situ (IAGOS aircraft, SHADOZ ozone sondes) measurements and six satellite products. To increase data coverage, the IAGOS and SHADOZ datasets are fused together over three regions (tropical Americas, Southeast Asia, and Malaysia/Indonesia). A major contribution is the use of satellite products with larger sample sizes to demonstrate strong sensitivity, including in the sign, of the trends detected by sparse sampling. The tropical tropospheric ozone distributions, trends, and statistics reported in the manuscript are important benchmarks of tropical tropospheric ozone for comparison with past and future trends, and for model evaluation. The paper should be suitable for publication in ACP once the major concerns noted below are addressed; other comments are intended to streamline the paper to ensure key messages are clearly communicated.

**We would like to thank the reviewer for this nice summary and we addressed their concerns below in bold text.**

Major Concerns

1. This assumption underlying the correction of satellite data with the in situ datasets from 2014-2019 at first seems in conflict with the conclusion that the in situ datasets are too sparse to reliably detect trends (for example lines 53-59; 769-771). Is the sample size sufficiently larger in this time period? Figure 8 is pointed to as the justification (lines 945-946) yet only shows trends in the satellite datasets, rather than the 2014-2019 mean in situ vs satellite data comparisons used for the correction (Figures 4 and 5) but the discussion of these figures is confusing as noted below, and it's not clear if the satellite instrument sensitivity is accounted for (see also point 4 below and comments on Figures 4 and 5). Clarification is needed to ensure the reader can understand the assumptions involved.

   **We use a very different approach for bias-correcting the satellite data than we use for trend estimation. We chose the 2014-2019 timeframe so that we could apply as many in situ data sets as possible , such as those from the ATom aircraft campaign. When we correct satellite data, we're focusing on the absolute ozone values, not their temporal variations. By analyzing five years of data from three different sources, we enhance the reliability of our findings due to a much larger sample size, as opposed to relying on a single dataset.**

2. The drift correction to OMI/MLS TCO is noted in Lines 479-480 and discussed in Section S2 but the actual documentation is quite thin if this manuscript is to serve as the sole

description. If it is documented elsewhere, references are needed. Where can the reader see the evidence for the drift discussed in lines 139-141? Why is there sufficient confidence to use the ozone sondes to correct the satellite drift in light of the issue so clearly demonstrated with trend detection in sparsely sampled datasets? The ground-based total ozone comparisons probably offer some critically needed independent evidence, assuming those do not suffer from the same sparse sampling as the sondes, yet rely on a personal communication. Why not include figures clearly demonstrating the drift in the supplement? Is the OMI CCD product sufficiently independent from the OMI/MLS to be used to cross-check? It is important to walk the reader through the evidence here.

**We have completed Section S4 "The OMI/MLS measurements and drift corrections" with a detailed demonstration of the drift correction with new figures included.**

3. Data Availability. Will the authors provide the fused regional datasets and trends, as well as their codes for deriving them? That would help to ensure reproducibility while also lowering the barrier for their use in model evaluation. Are all values included for all days with data in each year when determining quantiles or is some effort made to normalize sample sizes across months? Section 2.5 does not provide sufficient detail to allow reproducibility of the trend analysis. The coefficients fit to equation 1 could be reported in the supplement.

**We now provide the fused datasets and trends (including uncertainty and p-value associated with trend estimate, and fitted coefficients for ENSO/QBO in equation [1]) in Gaudel et al. (2024,** https://csl.noaa.gov/groups/csl4/modeldata/**).**

**Thanks for pointing out the issue of quantile quantification, our previous description of quantile/median trends is not accurate. We do not need to calculate the median data values in advance to apply median regression, as a median trend can be directly determined through all the individual data points. This is the major advantage of applying quantile regression (see Figure 1 of Chang et al. (2023) as a demonstration, also shown below). We thus revised the text as follows (L. 463):**

*Due to limited available sample sizes, only median trends (i.e., an estimate of the trend based on the median change in data distribution) are reported in this study. It should be noted that quantile regression is specifically designed to evaluate the distributional changes (determined by all available profiles). Although trends in extreme percentiles are not considered due to insufficient samples, by focusing on the median changes, our trend estimates are expected to be more robust against extreme variability or less impacted by potential large sampling bias (due to imbalanced sampling). The R and Python codes for implementing quantile/median regression are provided in the TOAR statistical guidance note (Chang et al., 2023).*

**For the code availability and further details on the data fusion methodology, we provided additional information in Supplementary Section S2:**

*Section S2: Analysis steps for data fusion methodology*

*The features of our data fusion are (i) to consider systematic ozone variability across vertical profile time series, instead of treating observations at different pressure surfaces as a set of an independent time series. By taking account of the vertical correlation, the method produces a more consistent trend estimates vertically, and the uncertainty can be effectively reduced; and (ii) to use the inverse of the (monthly) squared standard error as the weight when combining different sources of data records, so a record with a higher sampling frequency and/or lower variability has a higher influence. Analysis steps for data fusion can be described as follows (also described in Chang et al.(2022)):*

*For each data set and pressure surface (at 10 hPa vertical resolution), the time series is deseasonalized into data anomaly series, by using four harmonic functions. Explicitly, the anomalies are calculate by*

$$y_d = y - a_0 - a_1 sin(Month \times \pi/6) - a_2 cos(Month \times \pi/6) - a_3 sin(Month \times \pi/3) - a_4 cos(Month \times \pi/3)$$

1. *where y is the ozone value, and $(a_0, \ldots, a_4)$ are coefficients to determine the seasonal cycle (see Chang et al. (2023) for implementation code).*
2. *For each data set and pressure surface, the anomaly series is standardized by dividing its standard deviation (i.e. $y_d/SD(y_d)$ indicated as normalized deviation or ND in the figure), so the magnitude of data variability is similar between different pressure surfaces. The rationale of the regression problem is to find the best fit such that the sum of residuals is minimized. In terms of ozone profile, ozone values in the upper troposphere are typically stronger than the lower troposphere. Under this condition (since we consider vertical variability altogether), the statistical model prioritizes the reduction of fitted errors in the upper troposphere over the lower troposphere. Standardization removes this prioritization, and makes each vertical layer equally important across the troposphere. This consideration enables small scale variability to be better resolved by the statistical model.*
3. *Different sources of data records are combined by their standardized anomalies, and implemented under the framework of generalized additive models (GAM) and R package mgcv (wood, 2017). Based on the data preparation from previous steps, the main syntax for GAM data integration in R can be demonstrated as follows:*

```
> head(dt)
    Year Month index Pressure site Ozone_ppbv Ozone_ppbv_SE Ozone_anomaly_ppbv Ozone_standardized
248 1994     8     8      200    1   52.04750      9.643967        -0.07382282       -0.006870425
249 1994     8     8      210    1   46.90250      8.614597        -3.09289957       -0.312953398
250 1994     8     8      220    1   47.93800      7.589113        -1.12045218       -0.118577743
251 1994     8     8      230    1   46.49067      6.762999        -2.28952312       -0.245373816
252 1994     8     8      240    1   45.06875      6.251207        -3.52660169       -0.370912261
253 1994     8     8      250    1   44.44625      6.059981        -3.58347891       -0.391296508
> tail(dt)
      Year Month index Pressure site Ozone_ppbv Ozone_ppbv_SE Ozone_anomaly_ppbv Ozone_standardized
43014 2019    12   312      950    2   19.80000      11.63750          0.7552521         0.11666760
43015 2019    12   312      960    2   19.43333      11.70272          0.9353689         0.14621273
43016 2019    12   312      970    2   18.91667      11.89038          1.2253181         0.18773597
43017 2019    12   312      980    2   17.90000      12.27958          0.6877533         0.10584048
43018 2019    12   312      990    2   16.30000      13.05189         -0.3003434        -0.04605116
42929 2019    12   312     1000    2   14.43333      14.07423         -1.5527083        -0.22962453
> weights=(1/dt$Ozone_ppbv_SE^2)
> model=gam(Ozone_standardized ~ factor(site) + s(index, Pressure, bs="ds", k=2000), data=dt, weights=weights)
```

*where 'index' is monthly index spanned over study period, and 'Pressure' is the observed pressure surface. This formulation indicates that (i) temporal and vertical variabilities are modeled jointly (instead of two independent terms, based on the generalized thin plate splines); and (ii) the model fit is weighted by data uncertainty. The results can then be extracted by using predict(model, type='terms').*

*It should be noted that in terms of data fusion, it is also interesting to study remaining variability from each individual dataset, e.g., after separating regional variations from data, the local influence can thus be better understood. Nevertheless, in this study the number of datasets to be integrated is too few (only two or three) to identify the remainder in each dataset (Gomes, 2022). If sufficient datasets are available (e.g., five data sources), one can replace the term factor(site) with a more sophisticated representation:*

*ti(index, Pressure, site, bs=c("ds","fs"),d=c(2,1),k=c(300,5))*

*(see Wood (2017) for further details), so the remaining variability in each data source can be properly characterized.*

4. *The model produces fitted/fused monthly time series at each pressure layer. Trends can be estimated after the fitted values are transformed back to the units of ppbv (reverse the standardization). Further implementation details for trend analysis can be referred to Chang et al. (2023).*

[Figure]

**Figure 1 of Chang et al. (2023). A demonstration of the difference between a range of trend methods (upper panel) and percentile trends derived from quantile regression (QR, lower panel), based on surface ozone anomalies measured at Mace Head, Ireland (see Chang et al. (2023) for further details).**

**Chang, K. L., Cooper, O. R., Gaudel, A., Allaart, M., Ancellet, G., Clark, H., ... & Torres, C. (2022). Impact of the COVID-19 economic downturn on tropospheric ozone trends: An uncertainty weighted data synthesis for quantifying regional anomalies above western North America and Europe. AGU Advances, 3(2), e2021AV000542.**

**Chang, K. L., Schultz, M. G., Koren, G., & Selke, N. (2023). Guidance note on best statistical practices for TOAR analyses.**

**Gomes, D. G. (2022). Should I use fixed effects or random effects when I have fewer than five levels of a grouping factor in a mixed-effects model?. PeerJ, 10, e12794.**

**Wood, S. N. (2017). Generalized additive models: an introduction with R. Chapman and hall/CRC.**

4. How different are the vertical sensitivities of the various satellite instruments used to derive tropospheric column ozone? Lines 720-721 note the diminished sensitivity in the boundary layer, but it isn't clear if these are accounted for when comparing the satellite columns with those sampled in situ prior to bias-correcting the satellite data. Has an averaging kernel been applied to the ozone sonde or IAGOS or fused data, for example as in Zhang et al. ACP 2010 (doi:10.5194/acp-10-4725-2010)?

**There are several approaches to compare satellite with in situ data, applying the averaging kernels to the in situ data is one of them. We chose to not apply averaging kernels because we would like the differences to precisely count for the difference in vertical sensitivity; i.e. how well can a satellite retrieve the full depth of the tropospheric column ozone?**

Specific comments

Figures 4 and 5 and surrounding discussion can be streamlined by only including the apples-to-apples comparisons in the main text that are used in the correction as a positive offset (rather than a "high bias") is expected in Figure 4 for the satellite TCO products that retrieve through the full depth of the tropical troposphere. The alternative comparisons could be moved to the supplement to demonstrate the importance of using the correct vertical top of the column. More generally, the use of the word "bias" should be carefully considered in the text, as its use for some of the panels in Figures 4 and 5 may mischaracterize the situation given the known difference in the tropospheric column thickness for in situ versus remotely sensed datasets.

**We agree with the reviewer that we should carefully use the word "bias". We switched to the word "offset" when the definition of the column is known to be different. We think that the demonstration of the importance of working with a consistent definition of the column across the datasets belongs in the main manuscript. It is currently a substantial activity across TOAR-II focus working groups.**

Figure 6. Consider adding shading (2-sigma range?) to show if the fused data is statistically significantly different? If there is sufficiently larger confidence in the fused data, why not focus on only that dataset in the text, and move the comparisons to supplement?

**We have now updated Figure 6 and added the comparison between the fused and IAGOS only trends to the supplement Figure S21.**

Lines 665-671. This discussion is difficulty to follow. Isn't the MOZAIC dataset now included in IAGOS? In any case, this could be shown in Figure 7.

**The MOZAIC data were incorporated into the IAGOS program and Blot et al. studied the continuity of the two programs which are now only known as IAGOS. We decided to remove the sentence because it does not serve the main message of the discussion.**

Figure 7. Consider limiting the y-axis range to just the tropics to enhance legibility. Why are the months listed in the title?

**Limiting the y-axis bring distortions in the arrows and the coastlines. For clarity we decided to show the global map as it does not seem to alter the message of the figure. We removed the months listed in the title.**

Table 1 reports important information; is there a summary figure, perhaps in the style of the Figure 3 or 7 maps that could better communicate the key points, and the table could move to supplement?

**We decided to keep the table to visualize the numbers for precision, as well as for easy and accessible reference to readers for future comparisons.**

Lines 897-898. Is the range over the averages of the different products, or are spatial differences also part of this range?

**We found this sentence to be misleading and we rephrased it to give the absolute variability of the tropical tropospheric ozone: "From space, the spatial distribution of tropical tropospheric column ozone (TTCO) ranges from 10 to 40 DU in the 20ºS-20ºN latitude band."**

Lines 899-901 seem misleading given the known mismatch of tropospheric column depth; why not just report the correctly matched columns?

**We agree and we revised the text as follows: "The definition of the tropospheric column plays an important role in assessing tropical tropospheric ozone. Satellite data with a higher upper limit overestimate tropical column ozone compared to in-situ data. Mean biases reach up to 9 DU for OMPS/MERRA2 when compared to IAGOS, ATom and SHADOZ. The bias is 0 for IASI/GOME2 for which the column definition matches the in situ observations. – Section 3.3, Figure 4"**

Line 905. Need to define TTOB.

**Done**

Lines 908-909. What is being compared here? The equator versus mid-latitudes, or the parallel tropical latitude bands in the northern vs southern hemisphere?

**Both equator vs higher tropical latitude bands and tropical northern vs southern hemisphere are compared.**

Line 953-954. Some platforms have different vertical extents; isn't it just a matter of accounting for these differences rather than needing to define a common top?

**We agree with the reviewer: the considered satellite tropospheric ozone measurement techniques inherently lead to different vertical extents, so the data record should be harmonized to account for these differences. This is now clarified in the manuscript.**

Lines 954-957. Can this be expanded? What is the target here, to stitch together different satellite products or something else?

**We expanded the text as follow: "Additionally, there is a pressing need for the availability of common or joint and harmonized retrievals for satellite data from different platforms and instrument techniques, such as those provided by initiatives like TROPESS (TRopospheric Ozone and its Precursors from Earth System Sounding, https://tes.jpl.nasa.gov/tropess/)."**

Section S1 line 37. Figure S1 --> Figure S2?

**Corrected. Thank you.**

Section S1 lines 66-72. Is it worth noting that natural variability can influence trend attribution (not just detection)? Is the statement that sampling frequency versus natural variability are typically inseparable true? The authors seem to have succeeded in isolating sampling frequency issues by sampling the satellite data with the frequency of the in situ data.

**It is true that the influences of sampling frequency and natural variability are typically inseparable. These influences can be properly investigated only if we have (nearly) full sampled time series. Our approach to sample the satellite retrievals based on in-situ data availability can substantially reduce the impact of sampling uncertainty. Nevertheless, they are still not considered to be temporally and spatially collocated (we use satellite monthly values over a coarse grid box, not the exact sampling location and dates). Therefore, we are not able to conclude that we have succeeded in isolating the sampling impact.**

**In our separate work (Chang et al. (2024), ACP accepted, https://egusphere.copernicus.org/preprints/2024/egusphere-2023-2739/), we used the daily nighttime surface ozone measurements at Mauna Loa, Hawaii (1980-2021, representative of the lower free troposphere), to study the trend detection and attribution under different sampling frequencies. Under the circumstance when a (nearly) complete**

time series is available, we are able to isolate sampling impact and find that large trend uncertainty can be attributed to collocated meteorological variability. Unfortunately, limited data availability in the Tropics prohibits us from making such strong statements regarding sampling impact at the locations studied in the current sumission.

**We revised the text in Section S1 (l66) as:**

**"It should be noted that natural variability also plays a role in trend detection and attribution."**

**Chang, K.-L., Cooper, O. R., Gaudel, A., Petropavlovskikh, I., Effertz, P., Morris, G., and McDonald, B. C.: Technical note: Challenges of detecting free tropospheric ozone trends in a sparsely sampled environment, EGUsphere [preprint], https://doi.org/10.5194/egusphere-2023-2739, 2024.**

Section S3.  It seems that something is missing here; is it possible to have high confidence that there is not a trend?  Specifically for the Americas, should there instead be moderate confidence of an insignificant trend?  Should Malaysia/Indonesia be downgraded to low confidence?  The justifications here seem a bit arbitrary.

**The justifications we use to characterize the trends are by definition subjective. We chose the language to express confidence that there is a trend rather than that there is not a trend. The p-value, data gaps and sampling rate are all taken into consideration to rank the trends instead of simply using the p-value to judge if there is a trend or not, because the p-value does not account for the uncertainty in the monthly mean ozone values, which can be very uncertain in these data sets which have low sampling frequencies (Chang et al., 2024). In other words, the p-value alone reflects the true trend uncertainty only if all the monthly values are representative and complete. We can only have high confidence that there is not a trend in situations when we have a very high sampling frequency that yields accurate monthly mean ozone values.  As our monthly means are subject to large sampling biases, we cannot have high confidence that there is no trend.**

Figure S2: blue crosses --> diamonds?

**Corrected**

Figure S7: Caption/labels could better explain what is being plotted (i.e., why are there two panels per site; how is ND calculated). Consider plotting the 4 panels that should be compared directly on the same page.

**We updated the figures S12-14 improving the caption and the titles of the panels for clarity. We added a section in the supplement with more information on ND:** *Section S2. Analysis steps for data fusion methodology*

Figure S18: Is ECCAD using GFED fire emissions or something else? A reference or description would be helpful for readers not familiar with this inventory.
**ECCAD is not an inventory. It is a database and we added the reference Darras et al. (2018) that describe the database. In Figure S18 (now Figure S23), we show monthly CO from the GFED4 fire emissions inventory.**
*Darras S., Granier C., Liousse C., Boulanger D., Elguindi N., et al.. 2018. The ECCAD database, version 2: Emissions of Atmospheric Compounds & Compilation of Ancillary Data.*

**Citation**: https://doi.org/10.5194/egusphere-2023-3095-RC1

**Review 2**

**Summary**

This paper presents a comprehensive analysis of tropospheric ozone in the tropics. In situ data from IAGOS and SHADOZ are presented alongside a fused in situ product and long-term satellite records. Trends are presented for multiple regions in the tropics and analyzed using techniques such as satellite+in situ comparisons and bias corrections. The authors discuss contributing factors, such as tropopause definition, ozone precursor data, and data availability. Many previous studies are cited and presented clearly throughout the Introduction and Results. This paper will be a productive contribution to ACP once a couple of comments are addressed.

**We would like to thank the reviewer for this nice summary and we addressed their comments below in bold text.**

**Major comments**

- Section 2.1.1 states "To compare with the satellite data, the profiles were averaged monthly before being converted to a tropospheric column value ranging from the surface up to 270 hPa or up to the maximum altitude (~ 200 hPa)". A similar statement is made in Section 2.1.2 about the SHADOZ data. Did you consider applying satellite averaging kernels and priors to the IAGOS

and SHADOZ profiles in order to make a more direct comparison? Why did you decide against using this method?

**Applying an averaging kernel is an approach that degrades the vertical resolution of in situ data to match the vertical sensitivity of the satellite. It simplifies the comparison by highlighting the vertical sensitivity parameter as the cause of the differences. When satellite data provide the tropospheric column ozone, they are not truly the entire tropospheric column of ozone as they miss the information from the boundary layer. In this study, we would like to quantify the "real" difference between in situ data and satellite data at retrieving the tropospheric ozone column. We would like to count for all the possible reasons behind the differences: vertical sensitivity, top limit of the column, horizontal resolution, etc…**

- Section 2.3.4 states "This produced a 1-1.5 DU difference between the earlier and latter record for stratospheric column ozone, which prevents accurate trend detection from either MERRA2 stratospheric column ozone or the derived tropospheric column ozone from OMPS/MERRA2". However, the OMPS trend was reported in Figure 8. Why was OMPS included if an accurate trend detection is not possible?

  **In Figure 8, we are only showing the timeseries of ozone in Tg, to discuss comparisons between the satellite before and after adjustment in terms of ozone burden focusing on the seasonal and interannual variability. We do not report any trends (Tg/decade or ppbv/decade) from OMPS in the paper. We only report trends from OMI and OMI/MLS (Table 2).**

- A bullet point in the Conclusions section should be added to address the importance of tropospheric column definition, given your results showing the impact of columns calculated up to 270, 200, 150, or 100 hPa (in addition to your mention of this topic in the paragraph beginning on line 953).

  **We changed the 4th bullet point in the section "Present-day distribution" of the conclusion to "The definition of the tropospheric column plays an important role in assessing tropical tropospheric ozone. Satellite data with a higher upper limit overestimate tropical column ozone compared to in-situ data. Mean biases reach up to 9 DU for OMPS/MERRA2 when compared to IAGOS, ATom and SHADOZ. The bias is 0 for IASI/GOME2 for which the column definition matches the in situ observations."**

**Minor comments**

1. Line 277: define a.s.l.

   **Corrected**

2. Section 2.3.2 states that the tropospheric column "is constrained in the 15˚S-15˚N latitude band inherent to the CCD technique". How can the TROPOMI CCD method reach outside of this band, from 20°S to 20°N?

In principle, all CCD measurements from TROPOMI, OMI and TOMS should cover the 20°S-20°N latitude band but for the first version of this study we used the OMI CCD version that expands until 2019 and is more constrained in latitude. We now updated the OMI CCD so that it covers 20°S-20°N latitude band and expands until 2021. We updated Figure 8 and Table 2 accordingly, as well as the text in section 3.6 and in the conclusion.

3. OMI/MLS and OMPS both use the thermal tropopause, which varies seasonally. Did you consider recalculating the TTCO using a constant tropopause (at 100 hPa) to be more similar to OMI CCD? I suspect that the difference would be small, and this is not a critical edit to make, but it could be an interesting figure in the SI to present TTCO trends using OMI/MLS and OMPS up to the thermal tropopause versus up to 100 hPa.

This is a great point. The harmonization of the upper limit for the definition of the tropospheric column across the satellite products is a current activity from TOAR-II and will be the subject of one of the stand-alone papers from the report, which should be published in a year from now.

4. Check the grammar of the sentence spanning lines 388-389.

Corrected

5. Why are the CrIS and IASI/GOME2 tropopause pressures not plotted in Figure S1?

We haven't had access to the CrIS and IASI/GOME2 tropopause pressures from the groups and we produced the figure with the current information we had access to.

6. Define ND in Figures S7-S11.

We add a new section in the supplement: *Section S2. Analysis steps for data fusion methodology*

7. Many of the figures (e.g., Figures 2 and 3) use TCO in the figure labeling, but TTCO in the captions. Should they all be TTCO?

Yes, they should all be TTCO and we have corrected the figures. We corrected the caption of Figure 2 to spell out TTO as tropical tropospheric ozone because this figure shows profiles instead of columns.

8. Why does Figure S16 say "(nmol mol$^{-1}$ which is equivalent to nmol mol$^{-1}$)"?

We corrected the caption to "(nmol mol$^{-1}$ which is equivalent to ppbv)"

9. Line 699: change "cannow" to "can now".

Corrected

10. The Conclusion is presented in a straightforward and helpful manner. For readers who may only read the Conclusion of the paper, you may consider also listing the relevant sections/figures/tables associated with each bullet point so that those readers can easily find the sections that they're interested in learning more about.

**We added the section and figure numbers associated to the bullet points.**

11. Do you plan to extend this work to a global study? If so, please mention that in the Conclusion.

**One of the focus working group of the Tropospheric Ozone Assessment Report (TOAR) phase II titled HEGIFTOM is applying similar trends analysis globally using IAGOS, Ozonesondes, FTIR, and Brewer/Dobson (Umkehr) data and this is the subject of a publication in preparation for present TOAR special issue. We added a sentence at the end of the conclusion: "It is worth noting that similar trends analysis are currently applied at global scale using IAGOS, Ozonesondes, FTIR, and Brewer/Dobson (Umkehr) data."**

12. Why is Appendix A an appendix and not a section in the SI?

**Appendixes are directly embedded in the main manuscript and it is important that the tool we use to interpret the trends is directly accessible by the readers because it supports the main message of the manuscript.**

**Citation**: https://doi.org/10.5194/egusphere-2023-3095-RC2

---

## Author Comment (AC2)

**Response to the Community Comment by Rodrigo J. Seguel**

Comments by Rodrigo J. Seguel on behalf of the TOAR-II Steering Committee on:
Tropical tropospheric ozone distribution and trends from in situ and satellite data
Audrey Gaudel (corresponding author), Ilann Bourgeois, Meng Li, Kai-Lan Chang, Jerald
Ziemke, Bastien Sauvage, Ryan M. Stauffer, Anne M. Thompson, Debra E.
Kollonige, Nadia Smith, Daan Hubert, Arno Keppens, Juan Cuesta, Klaus-Peter Heue,
Pepijn Veefkind, Kenneth Aikin, Jeff Peischl, Chelsea R. Thompson, Thomas B.
Ryerson, Gregory J. Frost, Brian C. McDonald, Owen R. Cooper
This manuscript was submitted to ACP as part of the TOAR-II Community Special Issue
https://doi.org/10.5194/egusphere-2023-3095
Discussion started: 22 January 2024; discussion closes 18 April 2024
This review is by Rodrigo Seguel, member of the TOAR-II Steering Committee. The
primary purpose of these reviews is to identify any discrepancies across the TOAR-II
submissions, and to allow the author teams time to address the discrepancies. Additional
comments may be included with the reviews.
While members of the TOAR Steering Committee may post open comments on papers
submitted to the TOAR-II Community Special Issue, they are not involved with the
decision to accept or reject a paper for publication, which is entirely handled by the
journal's editorial team.

**We would like to thank Dr. Rodrigo Seguel for this comprehensive summary of the paper.
Our responses to his comments are found below in bold text.**

General comments
The authors have assessed the distribution and trends of tropical tropospheric ozone using
available ozone profiles measured by a suite of in situ instruments (IAGOS commercial
aircraft, the SHADOZ network, and the ATom aircraft campaign) and six satellite records
of tropical tropospheric column ozone (TROPOMI, OMI, OMI/MLS, OMPS/MERRA2,
CrIS, and IASI/GOME2). The authors have performed a great job of enhancing the trend
detectability and comparability of different data sources and also provided a thorough
discussion about bias between satellite and in situ data.
In particular, given the sparsity of the in-situ sampling over the tropics (time and space),
SHADOZ and IAGOS measurements were fused over some regions to enhance the trend
detection, which is based on quantile regression, as suggested by TOAR-II guidelines
(TOAR-II Recommendations for Statistical Analysis). Also, the ozone profiles from in
situ observations were converted to columns to evaluate the satellite products and adjust
the satellite biases, thus allowing the reduction of the satellite differences in the tropical
tropospheric ozone burden (TTOB).
Overall, the findings are consistent with the papers from TOAR-I and the papers
submitted to the TOAR-II Community Special Issue. In this regard, the low ozone levels
found over the Americas are consistent with the relatively low ozone mixing ratios
measured at ground level in the South American tropics compared with South American
extratropics values reported in: https://doi.org/10.5194/egusphere-2024-328

Minor comments
In section 2 (Methods), the authors clearly define the latitudinal band corresponding to the tropics for this study, which also follows the TOAR-II recommendation (TOAR-II Community Special Issue Guidelines). However, in lines **124** and **132**, they refer to the tropics between 30º S and 30º N, which includes the subtropics. For consistency, I suggest changing "tropics" to "tropics and subtropics," similar to line **101**.
**Thank you for this comment. We updated the lines 124 and 132 accordingly.**

Line 215-2016 (Caption of Figure 1): Please change "Africa, South Asia" to "Western Africa, India"
**Changed**

Line 257: Do you mean the disagreement is within ± 2%?
**We rephrased the sentence as follow: "*In comparisons of the reprocessed data with collocated total ozone spectrometers and satellite overpasses, the reprocessed SHADOZ total ozone column (TOC) disagreed with the independent data within 2% (Thompson et al., 2017).*"**

Line 283: Please check the parenthesis.
**The parenthesis are correct.**

Line 699: Change "cannow" to "can now"
**Changed**

Line 894-896 (Conclusions): In addition to the processes described by Kley et al. (1996), are there any other relevant processes that explain the low ozone values found in the Americas and the tropical South Pacific that may be suitable to discuss?

**The low ozone values in the lower troposphere measured above the Americas are mostly observed in the measurements above San Cristobal from the SHADOZ network and the Americas IAGOS region is mostly driven by measurements above Caracas (Venezuela) and Bogota (Columbia).**

**Oltmans et al. (1999) suggest that the lower values of ozone above San Cristobal may be explained by both strong ozone sink in the vicinity of the marine boundary layer as well as convection, the same mechanisms as above the Western Pacific and described in Kley et al. (1996). Above Caracas, low ozone levels close to the surface were already reported in Yamasoe et al. (2015) using IAGOS data. They show that except in the March-April-May season, most of the sources of ozone in Caracas are local. Sanhueza et al. (1999) show low levels of surface ozone in the south of Caracas during the wet season (May-December). Seguel et al. (2024) report lower ozone exposure (MDA8 health metric) in Bogota and Quito than other South American sites, which could be explained by intense vertical mixing as observed in Quito (Cazorla et al., 2021a; Cazorla, 2017).**

**We added this discussion in the manuscript as follow:**

*"Ozone levels are lowest above the tropical South Pacific (dark blue lines on the SHADOZ and ATom panels of Figure 2) and the Americas (IAGOS: mostly represented by measurements above Caracas and Bogota, and SHADOZ: San Cristobal, purple lines on both panels of Figure 2), with the 5th percentile below 10 nmol mol-1, particularly in the lower troposphere. These low ozone values are due to the ozone sink near the marine boundary layer coupled with deep convection above the tropical South Pacific (Kley et al., 1996) and San Cristobal (Oltmans et al., 1999). Above Caracas, the local influence is notable, with low ozone levels observed during the wet season (May-December) (Yamasoe et al, 2015; Sanhueza et al., 1999). Additionally, Seguel et al. (2024) report lower ozone exposure (MDA8 health metric) in Bogota and Quito than in other South American sites, likely due to intense vertical mixing as observed in Quito (Cazorla et al., 2021a; Cazorla, 2017)."*